# Multi-Objective One-Shot Pruning
# for Large Language Models

**Weiyu Chen**[1]   **Hansi Yang**[1]   **Yunhao Gou**[1,2]   **Han Shi**[3]   **Enliang Hu**[4]
**Zhenguo Li**[3]   **James T. Kwok**[1]
[1]The Hong Kong University of Science and Technology
[2]Southern University of Science and Technology
[3]Huawei Noah's Ark Lab
[4]Yunnan Normal University

## Abstract

Large Language Models (LLMs) have demonstrated remarkable capabilities across various tasks but require substantial computational resources, limiting their deployment in resource-constrained environments. While one-shot pruning methods can reduce model size without expensive retraining, they typically optimize for single objectives, ignoring LLMs' multi-faceted applications. We introduce Multi-Objective One-Shot Pruning (MOSP), which formulates LLM pruning as a multi-objective optimization problem. MOSP efficiently generates a Pareto set of pruned models representing different capability trade-offs, allowing users to select solutions aligned with their preferences. The proposed approach identifies share core support while enabling specialized support. Experiments across various LLMs and sparsity levels demonstrate MOSP's superior performance in navigating multi-objective trade-offs compared to baseline methods.

## 1  Introduction

Large Language Models (LLMs) have demonstrated exceptional performance across many scenarios, such as question answering, coding, and reasoning. However, their effectiveness typically requires massive model sizes and intensive computing resources, creating barriers for deployment in limited-resource settings. Network pruning [22, 17], which aims to remove redundant parameters, offers an important approach to create models with smaller computation cost while maintaining functionality.

For LLMs, traditional pruning methods [22, 17, 27, 19, 3] that involve iterative fine-tuning and/or extensive retraining are often prohibitively expensive due to the sheer scale of these models. Consequently, one-shot pruning techniques, which identify and remove weights without requiring retraining, have gained prominence [15, 37, 30]. These methods typically aim to minimize a layer-wise reconstruction error based on calibration data from a single, general-purpose dataset.

While effective in reducing model size, this single-objective focus overlooks the multi-faceted nature of modern LLMs, which are increasingly evaluated across diverse tasks like text comprehension, mathematical reasoning, and code generation. Users prioritize these capabilities differently, requiring models that are not only sparse but also adaptable to specific preferences. Current one-shot pruning methods lack mechanisms to address such multi-objective demands or tailor models to varying priorities. As discussed in Section 3.1, simple adaptations, like concatenating activation data or independently pruning and merging, often result in suboptimal performance or limited flexibility.

To address this gap, we introduce Multi-Objective One-Shot Pruning (MOSP), a novel framework designed to efficiently prune LLMs while considering multiple objectives simultaneously. Instead of producing a single pruned model, MOSP generates a Pareto set of solutions, where each solution

represents a different trade-off among the objectives. This allows users to select a pruned model that best aligns with their particular needs and preferences. MOSP achieves this through a multi-stage process: First, we identify a common core support of weights crucial across all tasks using Dual ADMM optimization. This involves formulating the problem as bilevel optimization with ADMM applied to both inner and outer levels, with proven convergence guarantees. Second, using the identified supports, we perform a simplified ADMM for each task separately. This decoupling of shared knowledge preservation from task-specific optimization enables efficient on-the-fly generation of specialized sparse models based on user-defined preference vectors, allowing effective exploration of the Pareto front.

The main contributions of this paper are as follows:

- We frame LLM pruning as a multi-objective optimization problem, explicitly addressing diverse user preferences, which is a novel perspective in LLM pruning.
- We introduce MOSP, an efficient one-shot pruning approach that generates a Pareto set of pruned LLMs, enabling flexible trade-offs across objectives.
- We provide a proof of convergence for the proposed dual ADMM method, which is non-trivial.
- Extensive experiments on various LLMs and sparsity levels show that MOSP outperforms baselines in navigating multi-objective trade-offs and provides a superior set of pruned models.

## 2  Background

### 2.1  Network Pruning

Network pruning reduces model complexity by eliminating redundant parameters, offering benefits like smaller size and faster inference [22, 17, 27, 19, 3]. Traditional methods, often computational expensive and/or requiring extensive retraining, are not suitable for Large Language Models (LLMs) due to their immense scale [15, 37]. Consequently, one-shot pruning techniques, which avoid retraining, are gaining prominence for LLMs.

LLM pruning methods can be categorized by the granularity of weights removed. Structured pruning removes entire components like neurons or attention heads [28, 1, 40, 42, 2, 11], maintaining regularity for hardware acceleration but usually leads to worse performance without retraining. Unstructured pruning removes individual weights [15, 37, 44, 41, 45], offering fine-grained control but creating irregular sparse patterns that can be harder to accelerate. Semi-structured pruning, such as N:M sparsity (N out of M weights kept), offers a compromise, balancing performance with hardware efficiency [46, 20]. Most unstructured LLM pruning methods are also applicable to semi-structured pruning. This paper focuses on one-shot unstructured and semi-structured LLM pruning.

### 2.2  One-Shot Unstructured and Semi-structured LLM Pruning

One-shot unstructured and semi-structured pruning methods aim to efficiently create sparse LLMs without retraining. These methods often optimize a layer-wise reconstruction error. Given an original dense weight matrix $\widehat{\boldsymbol{W}} \in \mathbb{R}^{c \times d}$ for a layer (where $c$ and $d$ are the input and output sizes, respectively), and calibration activations $\boldsymbol{X} \in \mathbb{R}^{nl \times c}$ (from $n$ samples each of length $l$), the goal is to find a sparse surrogate $\boldsymbol{W}$ by solving:

$$\min_{\boldsymbol{W} \in \mathbb{R}^{c \times d}} \left\| \boldsymbol{X}\widehat{\boldsymbol{W}} - \boldsymbol{X}\boldsymbol{W} \right\|_F^2 \quad \text{s.t.} \quad \|\boldsymbol{W}\|_0 \leq k, \tag{1}$$

where $\|\cdot\|_0$ is the $\ell_0$-norm, $\|\cdot\|_F$ is the Frobenius norm, and $k$ is the maximum number of non-zero elements.

Notable algorithms in this class include SparseGPT [15], which employs partial weight updates and adaptive mask selection to approximate the second-order information (Hessian) efficiently. Another approach, Wanda [37], offers simplicity by pruning weights based on the product of their magnitudes and the norms of corresponding input activations. ALPS [30] is an optimization-based framework that directly solves the $\ell_0$-constrained layer-wise reconstruction problem using the Alternating Direction Method of Multipliers (ADMM) [4, 10]. It identifies the optimal weight support and values, which are then refined by Preconditioned Conjugate Gradient (PCG) steps. However, these methods typically

consider calibration data $\boldsymbol{X}$ from a single, general-purpose dataset. This overlooks the fact that LLMs are often evaluated across multiple criteria. Different users may prioritize these objectives differently. Current one-shot pruning techniques generally do not address this need for customization, lacking mechanisms to generate models tailored to specific user preferences.

Another line of work considers the allocation of sparsity across different layers [41, 45, 23, 39]. Such layer-wise sparsity distribution strategies can often be combined with most of the aforementioned pruning algorithms to further improve performance. These approaches are orthogonal to the proposed methods and the two can be combined in a straightforward manner.

### 2.3 Multi-Objective Optimization

Multi-objective optimization (MOO) [31] optimizes $m$ objective functions simultaneously. Without loss of generality, we consider the minimization problem: $\min_{\boldsymbol{\theta} \in \boldsymbol{\Theta}} \mathbf{f}(\boldsymbol{\theta}) = \min_{\boldsymbol{\theta} \in \boldsymbol{\Theta}} (f_1(\boldsymbol{\theta}), \ldots, f_m(\boldsymbol{\theta}))$, where $\boldsymbol{\Theta}$ is the feasible decision space. A solution $\mathbf{a}$ dominates $\mathbf{b}$, denoted $\mathbf{a} \prec \mathbf{b}$, if $\forall i \in \{1, \ldots, m\} : f_i(\mathbf{a}) \leq f_i(\mathbf{b})$ and $\exists j \in \{1, \ldots, m\} : f_j(\mathbf{a}) < f_j(\mathbf{b})$. A feasible solution is Pareto-optimal when it is not dominated by any other feasible solution. The set of all Pareto-optimal decision vectors is called the Pareto set. The corresponding set of objective vectors, $\mathcal{F}^* = \{\mathbf{f}(\boldsymbol{\theta}) \mid \boldsymbol{\theta} \text{ is Pareto-optimal}\}$, is the Pareto front.

Gradient-based MOO methods have been widely adopted in deep learning [8]. They can be classified into three main categories: (i) Learning a single solution, with examples including MGDA [32, 14, 12], CAGrad [25], and Nash-MTL [33]; (ii) Learning a finite Pareto set, with examples including PMTL[24], EPO [29], MOO-SVGD [26], and GMOOAR [6]; and (3) Learning an infinite set of solutions, with examples including PHN [34], PaMaL [13], and LORPMAN [7].

All the aforementioned algorithms utilize gradient descent for optimization. However, the direct application of gradient descent is empirically ineffective in obtaining satisfactory solutions in the unstructured LLM pruning scenario, particularly when dealing with high sparsity ratios [30]. Consequently, these algorithms are not directly amenable to modification for one-shot LLM pruning.

## 3 Multi-Objective LLM Pruning

As mentioned in Section 2.2, current LLM pruning methods typically rely on calibration data $\boldsymbol{X}$ from a single, general-purpose dataset. However, LLMs are evaluated across multiple objectives, such as performance on general text, mathematical reasoning, and code generation. Single-objective approaches fail to generate models that satisfy users with varying preferences across these objectives. We therefore propose to formulate LLM pruning as a multi-objective optimization (MOO) problem.

Formally, let $\{\boldsymbol{X}^{(j)} \in \mathbb{R}^{n_j l_j \times c}\}_{j=1}^m$ be $m$ disjoint calibration sets, one for each objective $j$. Denote $f_j(\boldsymbol{W}) = \|\boldsymbol{X}^{(j)} \widehat{\boldsymbol{W}} - \boldsymbol{X}^{(j)} \boldsymbol{W}\|_F^2$. The problem is then a MOO problem:

$$\text{minimize} \quad \boldsymbol{f}(\boldsymbol{W}) := \big[ f_1(\boldsymbol{W}), \ldots, f_m(\boldsymbol{W}) \big]^\top \quad \text{s.t.} \quad \|\boldsymbol{W}\|_0 \leq k. \tag{2}$$

Relative importance of the objectives are represented by user preference $\boldsymbol{\lambda} = [\lambda_1, \ldots, \lambda_m]^\top \in \mathbb{R}_{\geq 0}^m$, where $\sum_{j=1}^m \lambda_j = 1$.

### 3.1 Straight-Forward Multi-Objective Extension of ALPS

We consider two straightforward multi-objective extensions of the SOTA single-objective LLM pruning method ALPS [30].[1] Experiments are performed on LLaMA-2-7B. For comparison, we show the performance (test perplexity) on applying ALPS to each task individually. While this per-task ALPS requires distinct models for different datasets, it serves as a performance upper bound.

**Extension 1: Activation Concatenation.** This stacks all calibration activations to form $\widetilde{\boldsymbol{X}} = [(\boldsymbol{X}^{(1)})^\top; \ldots; (\boldsymbol{X}^{(m)})^\top]^\top \in \mathbb{R}^{(\sum_j n_j l_j) \times c}$, and then apply ALPS to produce a pruned model.

*Limitation.* This extension produces only one single pruned model, which cannot adapt to varying user preferences across objectives. As can be seen from Table 1, dataset conflicts lead to performance

---

[1]We follow the parameter settings in ALPS.

Table 1: Test perplexities for multi-objective extensions of ALPS in Section 3.1, using LLaMA-2-7B at 70% sparsity (i.e., 70% of the weights are pruned).

|  | C4 | Code | GSM8K | Average |
|---|---|---|---|---|
| Per-task | 17.66 | 2.43 | 2.96 | 7.68 |
| Extension 1 | 20.43 | 2.63 | 3.12 | 8.72 |
| Extension 2 | 410.22 | 10.63 | 6.68 | 142.51 |

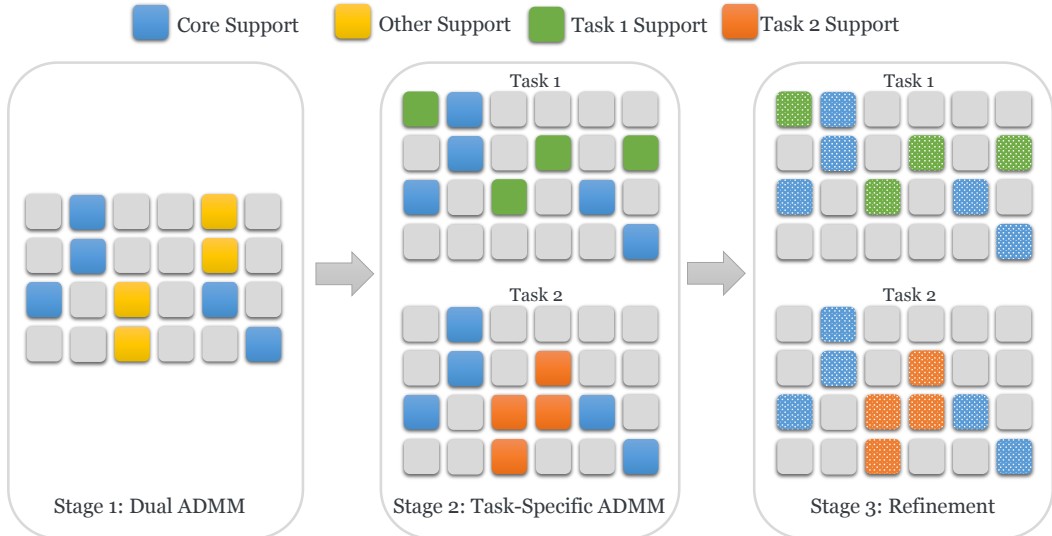

Figure 1: An overview of the proposed MOSP.

degradation compared to per-task pruning. While the $m$ calibration activations can be weighted by user preference, this requires running the full pruning process and storing separate sparse models for each preference vector $\boldsymbol{\lambda}^{(i)}$, incurring significant computational and storage costs.

**Extension 2: Independent Pruning then Merging.** This proceeds in three steps: (i) For each objective $j$, ALPS runs on $\boldsymbol{X}^{(j)}$ to obtain a sparse matrix $\boldsymbol{W}^{(j)}$. (ii) A merged matrix is computed as: $\boldsymbol{W}^{\text{merged}} = \sum_{j=1}^{m} \lambda_j \boldsymbol{W}^{(j)}$. (iii) Since $\boldsymbol{W}^{\text{merged}}$ is generally not $k$-sparse, it is re-pruned by retaining the $k$ entries with the largest magnitudes and setting all others to zero.

*Limitation.* As the individual supports $\text{Supp}(\boldsymbol{W}^{(j)})$'s can differ across tasks, the final re-pruning step may discard important weights, even if they are identified as important in the first step. This frequently leads to significant performance degradation across all objectives, as shown in Table 1.

## 3.2 Proposed Method

The limitations in Section 3.1 highlight the need for a new approach to efficiently generate pruned models tailored to diverse user preferences. In this section, we introduce Multi-Objective One-Shot Pruning (MOSP), which operates in three main stages. (i) Dual ADMM (Section 3.2.1), which uses a modified ADMM algorithm to jointly learn a primary sparse model, $\boldsymbol{W}$, and a core support, $\boldsymbol{W}_c$. (ii) Task-specific ADMM (Section 3.2.2), which performs a simplified ADMM procedure for each task $i$ separately, leveraging the supports identified in the first stage. (iii) Refinement, which further refines each model using Projected Conjugate Gradient (PCG) as in ALPS [30]. An overview of the proposed MOSP is illustrated in Figure 1. In the following, we will also discuss inference (Section 3.2.3), computational costs (Section 3.2.4), and the extension to N:M sparsity (Section 3.2.5).

### 3.2.1 Dual ADMM for Core Support Identification

We adapt ADMM [4, 10] to simultaneously learn two nested weight supports. We identify a primary weight matrix $\boldsymbol{W}$ with cardinality $k$, and a core weight matrix $\boldsymbol{W}_c$ inside $\boldsymbol{W}$ with cardinality $\alpha k$ (where $\alpha \in [0, 1]$). This is formulated as the following bi-level optimization problem:

$$\min_{\boldsymbol{W}_c \in \mathbb{R}^{c \times d}} \|\widetilde{\boldsymbol{X}}\widehat{\boldsymbol{W}} - \widetilde{\boldsymbol{X}}\boldsymbol{W}_c\|_F^2 + \gamma\|\widehat{\boldsymbol{W}} - \boldsymbol{W}_c\|_F^2, \|\boldsymbol{W}_c\|_0 \leq \alpha k, \text{Supp}(\boldsymbol{W}_c) \subset \text{Supp}(\boldsymbol{W}), \quad (3)$$

$$\text{subject to} \quad \boldsymbol{W} = \arg\min_{\boldsymbol{W}' \in \mathbb{R}^{c \times d}} \|\widetilde{\boldsymbol{X}}\widehat{\boldsymbol{W}} - \widetilde{\boldsymbol{X}}\boldsymbol{W}'\|_F^2 + \gamma\|\widehat{\boldsymbol{W}} - \boldsymbol{W}'\|_F^2, \|\boldsymbol{W}'\|_0 \leq k, \quad (4)$$

where $\widetilde{\boldsymbol{X}} = [(\boldsymbol{X}^{(1)})^\top; \ldots; (\boldsymbol{X}^{(m)})^\top]^\top$ is the concatenated activation, and $\gamma \geq 0$ is the regularization parameter.

To solve this optimization problem, we apply ADMM to both the inner and outer level problems. Besides the standard dual variables $\boldsymbol{V}$ and $\boldsymbol{V}_c$, ADMM also introduces auxiliary variables $\boldsymbol{D}$ and $\boldsymbol{D}_c$ for the sparsity constraints. We initialize $\boldsymbol{W}^{(0)}, \boldsymbol{D}^{(0)}, \boldsymbol{W}_c^{(0)}$, and $\boldsymbol{D}_c^{(0)}$ to $\widehat{\boldsymbol{W}}$, and set both $\boldsymbol{V}^{(0)}$ and $\boldsymbol{V}_c^{(0)}$ to $\boldsymbol{0}$. Let $\rho$ be the ADMM penalty parameter and $\boldsymbol{H} = (\widetilde{\boldsymbol{X}})^\top\widetilde{\boldsymbol{X}} + \gamma\boldsymbol{I}$. At iteration $t$, the updates are performed sequentially for the weight matrices, auxiliary variables, and dual variables as follows. The detailed derivation is in Appendix A.

First, the primary weight matrix $\boldsymbol{W}^{(t+1)}$ and the core weight matrix $\boldsymbol{W}_c^{(t+1)}$ are computed as:

$$\boldsymbol{W}^{(t+1)} = (\boldsymbol{H}+\rho\boldsymbol{I})^{-1}(\boldsymbol{H}\widehat{\boldsymbol{W}} - \boldsymbol{V}^{(t)} + \rho\boldsymbol{D}^{(t)}), \boldsymbol{W}_c^{(t+1)} = (\boldsymbol{H}+\rho\boldsymbol{I})^{-1}(\boldsymbol{H}\widehat{\boldsymbol{W}} - \boldsymbol{V}_c^{(t)} + \rho\boldsymbol{D}_c^{(t)}). \quad (5)$$

Next, $\boldsymbol{D}$ and $\boldsymbol{D}_c$ are updated. Let $\widetilde{\boldsymbol{W}}^{(t+1)} = \boldsymbol{W}^{(t+1)} + \boldsymbol{V}^{(t)}/\rho$ and $\widetilde{\boldsymbol{W}}_c^{(t+1)} = \boldsymbol{W}_c^{(t+1)} + \boldsymbol{V}_c^{(t)}/\rho$. The primary support projection yields $\boldsymbol{D}^{(t+1)} = P_k(\widetilde{\boldsymbol{W}}^{(t+1)})$ by selecting the $k$ largest-magnitude elements of $\widetilde{\boldsymbol{W}}^{(t+1)}$. The primary support is $\mathcal{S}^{(t+1)} = \text{Supp}(\boldsymbol{D}^{(t+1)})$. For the core support projection, $\widetilde{\boldsymbol{W}}_c^{(t+1)}$ is projected onto matrices with at most $\alpha k$ non-zero elements, constrained such that its support is a subset of $\mathcal{S}^{(t+1)}$. This is achieved by first masking $\widetilde{\boldsymbol{W}}_c^{(t+1)}$ with $\mathcal{S}^{(t+1)}$ and then selecting the $\alpha k$ largest-magnitude elements in this masked matrix: $\boldsymbol{D}_c^{(t+1)} = P_{\alpha k}(\widetilde{\boldsymbol{W}}_c^{(t+1)} \odot \mathcal{S}^{(t+1)})$. The core support is then $\mathcal{S}_c^{(t+1)} = \text{Supp}(\boldsymbol{D}_c^{(t+1)})$. Finally, the dual variables $\boldsymbol{V}$ and $\boldsymbol{V}_c$ are updated as:

$$\boldsymbol{V}^{(t+1)} = \boldsymbol{V}^{(t)} + \rho(\boldsymbol{W}^{(t+1)} - \boldsymbol{D}^{(t+1)}), \boldsymbol{V}_c^{(t+1)} = \boldsymbol{V}_c^{(t)} + \rho(\boldsymbol{W}_c^{(t+1)} - \boldsymbol{D}_c^{(t+1)}). \quad (6)$$

The above steps are repeated until convergence, outputting weight matrix $\boldsymbol{W}$ and core support $\mathcal{S}_c$.

**Convergence.** The following Theorem guarantees convergence of the algorithm. The proof is given in Appendix B.

**Theorem 1.** *Let $\{\mathbf{D}^{(t)}\}_{t=0}^\infty$, $\{\mathbf{W}^{(t)}\}_{t=0}^\infty$, $\{\mathbf{D}_c^{(t)}\}_{t=0}^\infty$ and $\{\mathbf{W}_c^{(t)}\}_{t=0}^\infty$ be the sequences generated by the algorithm in Section 3.2.1. Suppose the penalty parameter $\{\rho_t\}_{t=1}^\infty$ satisfies $\sum_{t=1}^\infty 1/\rho_t < \infty$. We have*

$$\max\{\|\mathbf{D}^{(t+1)} - \mathbf{D}^{(t)}\|_F, \|\mathbf{W}^{(t+1)} - \mathbf{D}^{(t+1)}\|_F, \|\mathbf{D}_c^{(t+1)} - \mathbf{D}_c^{(t)}\|_F, \|\mathbf{W}_c^{(t+1)} - \mathbf{D}_c^{(t+1)}\|_F\} \leq C/\rho_t,$$

*where $C$ is a constant depending on $\mathbf{X}$, $\widehat{\mathbf{W}}$, and $\sum_{t=1}^\infty 1/\rho_t$. In particular, there exists matrix $\bar{\mathbf{D}}$ and $\bar{\mathbf{D}}_c$ such that $\mathbf{D}^{(t)} \to \bar{\mathbf{D}}$, $\mathbf{W}^{(t)} \to \bar{\mathbf{D}}$, $\mathbf{D}_c^{(t)} \to \bar{\mathbf{D}}_c$, and $\mathbf{W}_c^{(t)} \to \bar{\mathbf{D}}_c$ as $t \to \infty$.*

### 3.2.2 Task-Specific ADMM

After identifying the core support $\mathcal{S}_c$ and a primary weight matrix $\boldsymbol{W}$ in Section 3.2.1, ADMM is applied individually to each task. This aims to obtain task-specific weights $\boldsymbol{W}_i$ ($i = 1, \ldots, m$) while preserving the core support $\mathcal{S}_c$. This can be formulated as the following optimization problem:

$$\min_{\boldsymbol{W}_i \in \mathbb{R}^{c \times d}} \|\boldsymbol{X}^{(i)}\widehat{\boldsymbol{W}} - \boldsymbol{X}\boldsymbol{W}_i\|_F^2 + \gamma\|\widehat{\boldsymbol{W}} - \boldsymbol{W}_i\|_F^2 \quad \text{s.t.} \quad \|\boldsymbol{W}_i\|_0 \leq k, \text{Supp}(\boldsymbol{W}_c) \subset \text{Supp}(\boldsymbol{W}_i). \quad (7)$$

For initialization, matrix $\boldsymbol{W}$ from Section 3.2.1 is used. As this initialization is strong, we use a fixed ADMM parameter $\rho$. This allows pre-computation of the Cholesky decomposition of $(\boldsymbol{H}_i + \rho\boldsymbol{I})$ (where $\boldsymbol{H}_i = (\boldsymbol{X}^{(i)})^\top\boldsymbol{X}^{(i)} + \gamma\boldsymbol{I}$) for each task $i$, significantly speeding up the optimization process.

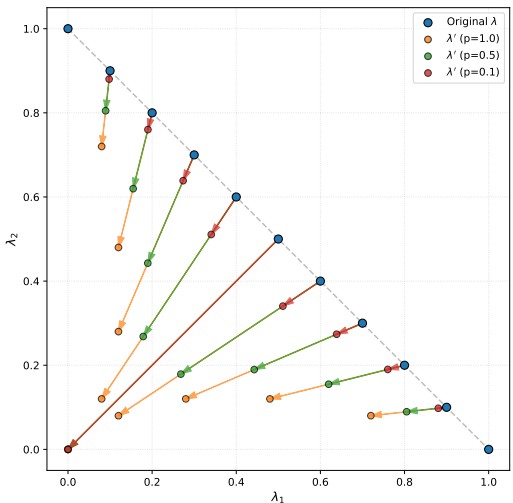

Figure 2: Example illustrating the mapping from $\boldsymbol{\lambda}$ to $\boldsymbol{\lambda}'$ for $m = 2$. Vectors closer to $(\frac{1}{2}, \frac{1}{2})$ shrink toward the origin, while vectors at the corners remain unchanged.

**Algorithm.** For each task $i$, ADMM sequentially updates $\boldsymbol{W}_i$, $\boldsymbol{D}_i$, and $\boldsymbol{V}_i$. At ADMM iteration $t$, weight $\boldsymbol{W}$ is updated as:

$$\boldsymbol{W}_i^{(t+1)} = (\boldsymbol{H}_i + \rho \boldsymbol{I})^{-1}(\boldsymbol{H}_i \widehat{\boldsymbol{W}} - \boldsymbol{V}_i^{(t)} + \rho \boldsymbol{D}_i^{(t)}).$$

Next, the auxiliary variable $\boldsymbol{D}_i$ is updated. Given $\widetilde{\boldsymbol{W}}_i^{(t+1)} = \boldsymbol{W}_i^{(t+1)} + \boldsymbol{V}_i^{(t)}/\rho$, the projection $P_{k,\mathcal{S}_c}(\cdot)$ first retains all elements of $\widetilde{\boldsymbol{W}}_i^{(t+1)}$ corresponding to $\mathcal{S}_c$. Then, it selects the $k - |\mathcal{S}_c|$ largest-magnitude elements outside $\mathcal{S}_c$. The resulting matrix $\boldsymbol{D}_i^{(t+1)} = P_{k,\mathcal{S}_c}(\widetilde{\boldsymbol{W}}_i^{(t+1)})$ comprises these selected elements from $\widetilde{\boldsymbol{W}}_i^{(t+1)}$, with all others set to zero. Finally, the dual variable is updated:

$$\boldsymbol{V}_i^{(t+1)} = \boldsymbol{V}_i^{(t)} + \rho(\boldsymbol{W}_i^{(t+1)} - \boldsymbol{D}_i^{(t+1)}).$$

For efficiency, the above steps are run for a maximum of 10 iterations, outputting $\{\boldsymbol{W}_i\}_{i=1}^m$.

### 3.2.3 Inference

During inference, given a user preference vector $\boldsymbol{\lambda} = [\lambda_1, \ldots, \lambda_m]^\top$, we adjust the impact of task-specific corrections $\{\boldsymbol{W}_i - \boldsymbol{W}\}$ and create a tailored model with weight matrix $\boldsymbol{W}_{\boldsymbol{\lambda}} = P_{k,\mathcal{S}_c}(\boldsymbol{W} + \sum_{i=1}^m \lambda_i'(\boldsymbol{W}_i - \boldsymbol{W}))$ on the fly. Here, $P_{k,\mathcal{S}_c}(\cdot)$ is a projection that ensures $\boldsymbol{W}_{\boldsymbol{\lambda}}$ has at most $k$ non-zero elements and includes all elements corresponding to the core support $\mathcal{S}_c$. We scale $\boldsymbol{\lambda}$ based on its $\ell_1$ distance from a uniform distribution, normalized by the maximum possible distance:

$$\boldsymbol{\lambda}' = \left( \frac{\sum_{j=1}^m |\lambda_j - \frac{1}{m}|}{2(1 - \frac{1}{m})} \right)^p \boldsymbol{\lambda},$$

where $p$ is a hyperparameter. For uniform $\boldsymbol{\lambda}$ (i.e., $\lambda_j = 1/m, \forall j$), $\sum_{i=1}^m \lambda_i'(\boldsymbol{W}_i - \boldsymbol{W}) = \boldsymbol{0}$ as the global weight $\boldsymbol{W}$ should provide a balanced solution. For a one-hot $\boldsymbol{\lambda}$, we have $\boldsymbol{\lambda}' = \boldsymbol{\lambda}$. Thus, the mapping from $\boldsymbol{\lambda}$ to $\boldsymbol{\lambda}'$ smoothly interpolates: near-uniform preferences favor the global model, while task-specific preferences amplify task-specific adjustments (Figure 2). Moreover, a larger $p$ delays task-specific adjustments until $\boldsymbol{\lambda}$ is highly skewed, while a smaller $p$ enables earlier transitions. In the experiments, we simply use $p = 0.5$.

### 3.2.4 Computation Cost

**Stage 1 (Section 3.2.1):** In this stage, the computational bottleneck is the calculation of $(\boldsymbol{H} + \rho \boldsymbol{I})^{-1}$. To address this, similar to ALPS [30], we pre-compute and store the eigen-decomposition of $\boldsymbol{H}$

as $QMQ^T$, where $Q$ is the matrix of eigenvectors and $M$ is the diagonal matrix of eigenvalues. In each iteration, the stored $Q$ and $M$ can then be reused to efficiently compute $(H + \rho I)^{-1}$ as $Q(M + \rho I)^{-1}Q^T$. Note that since the two optimization subproblems (for $W$ and $W_c$) use the same matrix $H$, this eigen-decomposition needs to be performed only once at the beginning of this stage, with a complexity $\mathcal{O}(c^3)$. The subsequent per-iteration computational complexity is $\mathcal{O}(c^2 d)$. Compared to ALPS, this stage introduces a minor overhead consisting of a few additional matrix multiplications and additions.

**Stage 2 (Section 3.2.2):** In this stage, we use a fixed $\rho$, which enables the pre-computation of the Cholesky decomposition of $(H_i + \rho I)$, followed by the computation of its inverse based on the decomposition, once for each task. While the Cholesky decomposition has $\mathcal{O}(c^3)$ complexity, it is significantly more computationally efficient in practice. After the inverse is computed, the per-iteration computational complexity for this stage becomes $\mathcal{O}(c^2 d)$. For $m$ tasks, the total per-iteration computational complexity is $\mathcal{O}(mc^2 d)$.

In Section 4.2, we will demonstrate that the computational overhead of the proposed method is small in comparison to ALPS.

### 3.2.5 Extension to Semi-Structured Sparsity

Similar to the other unstructured pruning techniques, the proposed method extends naturally to semi-structured sparsity by modifying the $D$-update projection steps.

**Stage 1 (Section 3.2.1):** First, partition $\widetilde{W}^{(t+1)}$ into non-overlapping blocks of $M$ elements. Retain the $N$ largest-magnitude elements in each block via projection $P_{N:M}(\cdot)$, yielding $D^{(t+1)} = P_{N:M}(\widetilde{W}^{(t+1)})$. To identify the core support, we compute the absolute sum of elements in each block of $\widetilde{W}_c^{(t+1)}$. Designate the $l$ blocks with smallest sums as "weak blocks," and define the core support $\mathcal{S}_c$ by excluding these blocks. The result is $D_c^{(t+1)} = P_{N:M}(\widetilde{W}_c^{(t+1)}) \odot \mathcal{S}_c$.

**Stage 2 (Section 3.2.2):** Using core support $\mathcal{S}_c$ from Stage 1, apply the projection $P_{N:M,\mathcal{S}_c}(\cdot)$ by preserving elements in $\widetilde{W}^{(t+1)}$ at positions in $\mathcal{S}_c$ while maintaining the N:M structure for the remaining elements. Specifically, retain up to $N$ largest-magnitude elements in each block of $M$ non-core blocks, resulting in $D^{(t+1)} = P_{N:M,\mathcal{S}_c}(\widetilde{W}^{(t+1)})$. This ensures the N:M sparsity pattern while preserving the identified core support.

## 4   Experiments

In this section, we show the experimental results of the proposed Multi-Objective One-Shot Pruning (MOSP). We begin by outlining the setup in Section 4.1. Subsequently, we demonstrate the effectiveness of MOSP on a two-objective pruning scenario (general language understanding and mathematical reasoning) in Section 4.2. We then extend the evaluation to three objectives, incorporating code generation capabilities, in Section 4.3. Finally, we provide ablation studies in Section 4.4.

### 4.1   Setup

We evaluate MOSP across multiple LLMs including Llama-2 [38], Llama-3 [16], and OPT series [43]. We consider three representative datasets to evaluate model performance across different domains: (1) General Text: C4 [36]. (2) Mathematical Reasoning: GSM8K [9]. (3) Code Generation: Code [5]. Following [15, 30], we use a calibration set of 128 segments (up to 2048 tokens) and evaluate using perplexity (PPL) on the test sets (the lower the better).

We compare MOSP with state-of-the-art unstructured pruning methods: Magnitude Pruning (MP) [18], Wanda [37], SparseGPT [15], and ALPS [30]. We use the same pruning datasets across all methods. For Wanda, SparseGPT, and ALPS, we use the activation concatenation strategy in Section 3.1. We exclude independent pruning then merging (extension 2) due to its poor performance (as shown in Table 1). The penalty $\rho$ is adapted using the same strategy as ALPS [30]. We set $\alpha = 0.5$ and $p = 0.5$ without hyperparameter tuning. Additional experimental details are provided in Appendix D.

## 4.2 Two-Objective Pruning

We first demonstrate the effectiveness of MOSP for pruning Llama-2-7B on two datasets: `C4` and `GSM8K`. The unpruned Llama-2-7B baseline achieves test PPL of 6.97 on `C4` and 2.73 on `GSM8K`.

Figure 3 shows the performance at 70% unstructured sparsity and 2:4 semi-structured sparsity. MP and Wanda are omitted due to their significantly inferior performance. As can be seen, the baseline algorithms produce a single pruned model, whereas MOSP generates a diverse Pareto set, demonstrating the trade-off between general language

Table 2: Comparison of time and GPU memory consumption between ALPS and the proposed MOSP.

| Method | Time | GPU Memory |
|--------|-------|------------|
| ALPS | 1.09h | 18.7GB |
| MOSP | 1.36h | 20.3GB |

understanding and mathematical reasoning. This allows users to select models that align with their preferences.

In Figure 4, we vary the sparsity ratio. As can be seen, MOSP generates meaningful Pareto fronts across varying sparsity settings. Notably, higher sparsity ratios amplify the trade-off region, emphasizing the value of multi-objective optimization at greater compression levels.

Table 2 examines the time and GPU memory overhead of the proposed MOSP compared with ALPS. As can be seen, both the time and memory overhead of the proposed method are small.

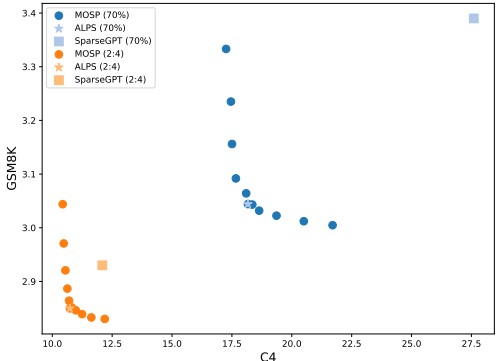

Figure 3: Test PPL on `C4` and `GSM8K` for Llama-2-7B pruned to 70% unstructured sparsity and 2:4 semi-structured sparsity.

Figure 4: Test PPL on `C4` and `GSM8K` for Llama-2-7B pruned to 50%, 60%, 70%, 80% unstructured sparsity and 2:4 semi-structured sparsity.

## 4.3 Three-Objective Pruning

In this section, we evaluate MOSP in a scenario involving three objectives: general language understanding (`C4`), mathematical reasoning (`GSM8K`), and code generation (`Code`). Figure 5 shows the solutions obtained by MOSP on pruning the Llama-2-7B model to 70% unstructured sparsity. Each point on the resulting three-dimensional surface represents a pruned model, generated using one of the 36 uniform preference vectors ($\lambda$). As can be seen, MOSP successfully identifies a diverse set of models spanning different objective trade-offs. More figures for different models pruned to different sparsity levels are provided in Appendix E.

Table 3 provides a quantitative analysis comparing models pruned using the baselines with representative points selected from the models obtained by MOSP for Llama-2-7B at 70% sparsity. These selected points correspond to different preference vectors: $\lambda^0$ represents balanced preference across all three objectives, while $\lambda^1$, $\lambda^2$, and $\lambda^3$ heavily favor `C4`, `GSM8K`, and `Code`, respectively, with $\lambda^4$ representing an intermediate preference. The results clearly demonstrate MOSP's ability to effectively navigate the trade-off spaces by adjusting the preference vector $\lambda$. For instance, $\lambda^1$ yields the lowest PPL on `C4`, while $\lambda^3$ achieves the lowest PPL on `Code`.

Table 4 compares the Hypervolume (HV) [47, 21] achieved by MOSP with baseline methods for pruning various OPT and Llama models to different sparsity targets. A higher HV value indicates a better approximation of the true Pareto front, reflecting solutions that are both high-performing

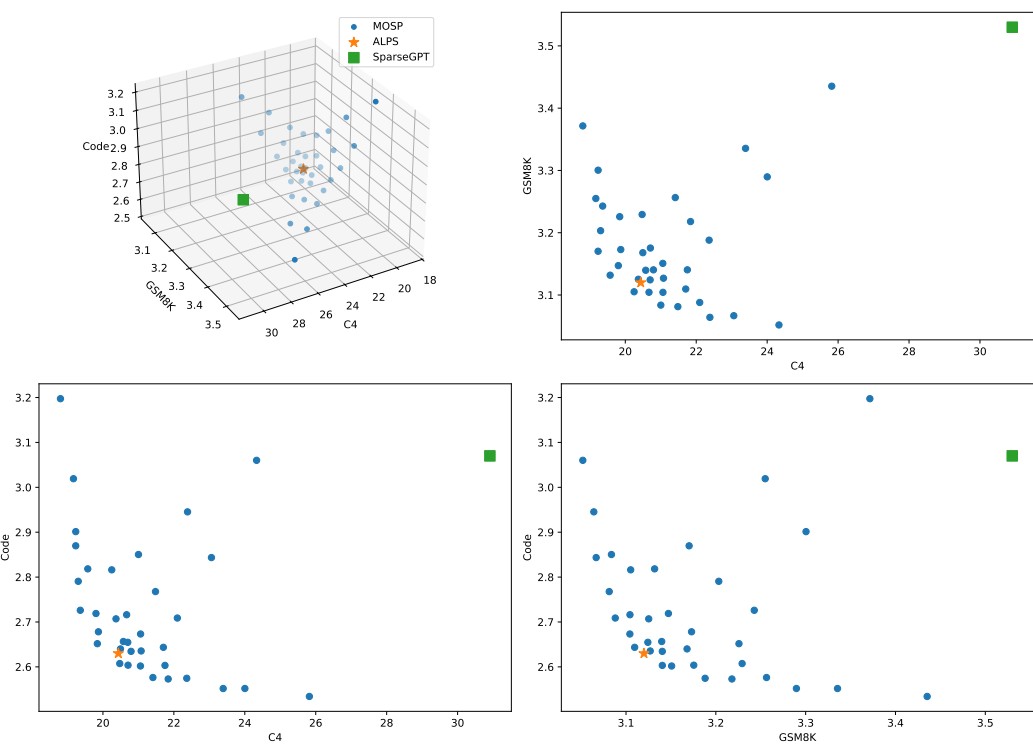

Figure 5: Test PPL on `C4`, `GSM8K`, and `Code` for Llama-2-7B pruned to 70% unstructured sparsity. The top left figure shows 3D view and the other three figures show different 2D projections.

Table 3: Test PPL of Llama-2-7B pruned to 70% unstructured sparsity with different preference vectors. $\boldsymbol{\lambda}^0 = [0.33, 0.33, 0.33], \boldsymbol{\lambda}^{(1)} = [1, 0, 0], \boldsymbol{\lambda}^{(2)} = [0, 1, 0], \boldsymbol{\lambda}^{(3)} = [0, 0, 1], \boldsymbol{\lambda}^{(4)} = [0.72, 0.14, 0.14]$.

| Method | C4 | GSM8K | Code |
|---|---|---|---|
| MP | 9839.28 | 6191.83 | 3285.59 |
| Wanda | 73.64 | 12.90 | 10.51 |
| SparseGPT | 30.91 | 3.53 | 3.07 |
| ALPS | 20.44 | 3.12 | 2.63 |
| MOSP ($\boldsymbol{\lambda}^0$) | 20.44 | 3.12 | 2.63 |
| MOSP ($\boldsymbol{\lambda}^1$) | 18.80 | 3.37 | 3.20 |
| MOSP ($\boldsymbol{\lambda}^2$) | 24.33 | 3.05 | 3.06 |
| MOSP ($\boldsymbol{\lambda}^3$) | 25.82 | 3.43 | 2.53 |
| MOSP ($\boldsymbol{\lambda}^4$) | 19.30 | 3.20 | 2.79 |

(closer to the ideal point) and diverse (covering a broader range of trade-offs). The reference point is set as 1.1 times the PPL of the model obtained using SparseGPT. Note that since the reference point varies across different sparsity levels and models, HV comparisons are only meaningful within the same model-sparsity combination. Results show that MOSP consistently outperforms SparseGPT and ALPS in HV. Wanda performs far below the reference point, yielding an HV of zero.

## 4.4 Ablation Studies

This section studies the impact of key components in MOSP: the support sharing parameter $\alpha$ and the preference mapping strategy. All experiments use Llama-2-7B pruned to 70% sparsity, with the hyperparameters kept consistent with the other experiments.

**Effect of $\alpha$.** Parameter $\alpha$ controls the degree of support sharing, interpolating between fully task-specific supports ($\alpha = 0$) to a completely shared support ($\alpha = 1$). Figure 6a shows that neither

Table 4: Comparison of HV on Llama-2, Llama-3, and OPT models across various sparsity levels

| Sparsity | Method | OPT-2.7B | Llama-2-7B | Llama-3-8B | Llama-2-13B | OPT-30B |
|---|---|---|---|---|---|---|
| 2:4 | Wanda | 0.00 | 0.00 | 0.00 | 0.00 | 0.00 |
| | SparseGPT | 0.39 | 0.09 | 0.27 | 0.07 | 0.16 |
| | ALPS | 0.99 | 0.38 | 1.39 | 0.21 | 0.28 |
| | MOSP | **1.18** | **0.48** | **1.70** | **0.25** | **0.31** |
| 70% | Wanda | 0.00 | 0.00 | 0.00 | 0.00 | 0.00 |
| | SparseGPT | 0.70 | 0.33 | 0.99 | 0.20 | 0.20 |
| | ALPS | 4.68 | 7.73 | 25.36 | 3.35 | 0.72 |
| | MOSP | **5.67** | **9.59** | **30.85** | **4.24** | **0.85** |
| 80% | Wanda | 0.00 | 0.00 | 0.00 | 0.00 | 0.00 |
| | SparseGPT | 4.86 | 4.83 | 9.28 | 2.02 | 0.90 |
| | ALPS | 168.50 | 770.74 | 1180.42 | 220.53 | 43.56 |
| | MOSP | **188.26** | **887.52** | **1287.07** | **259.21** | **48.65** |

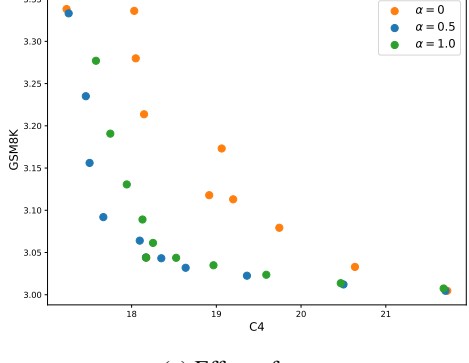

(a) Effect of $\alpha$ .

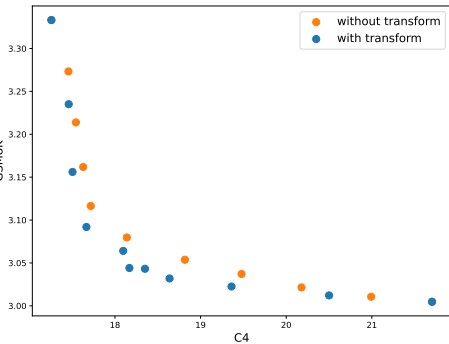

(b) Effect of preference transformation.

Figure 6: Ablation studies for Llama-2-7B pruned to 70% sparsity.

extreme yields the optimal trade-off. An intermediate $\alpha$, as used in our main experiments, allows partial overlap in supports. This enables MOSP to maintain similarity for better interpolated models while still permitting task-specific specialization.

**Effect of preference transformation.** Figure 6b shows the effect of the transformation from $\boldsymbol{\lambda}$ to $\boldsymbol{\lambda}'$. While the performance at the extreme ends remains the same, other models generally achieve better performance when the transformation is applied. This indicates that the performance transformation helps in finding superior intermediate models.

## 5 Conclusion

In this paper, we consider the multi-objective nature of LLM pruning and formulate it as a MOO problem. The proposed method MOSP, with proven convergence, efficiently identifies models with varying trade-offs across different objectives. Experiments on representative models demonstrate our approach's effectiveness in providing tailored models based on user preferences.

*Limitations.* We only evaluate MOSP on some representative models. Extending this evaluation to a broader range of LLMs would be valuable. Additionally, considering more objectives is an interesting future work.

## Acknowledgment

This research was supported in part by the Research Grants Council of the Hong Kong Special Administrative Region (Grants 16202523 and HKU C7004-22G).

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

# A  Detailed Derivation of Dual ADMM

We use the Alternating Direction Method of Multipliers (ADMM) to solve these constrained optimization problems. The updates for the primary weights $(\boldsymbol{W}, \boldsymbol{D}, \boldsymbol{V})$ and core weights $(\boldsymbol{W}_c, \boldsymbol{D}_c, \boldsymbol{V}_c)$ are performed sequentially within each iteration $t$. Let $\boldsymbol{H} = (\widetilde{\boldsymbol{X}})^\top \widetilde{\boldsymbol{X}} + \gamma \boldsymbol{I}$.

## A.1  ADMM for Inner Problem

The inner problem (4) aims to find the primary weight matrix $\boldsymbol{W}$. The problem is:

$$\min_{\boldsymbol{W}} \left\| \widetilde{\boldsymbol{X}} \widehat{\boldsymbol{W}} - \widetilde{\boldsymbol{X}} \boldsymbol{W} \right\|_F^2 + \gamma \left\| \widehat{\boldsymbol{W}} - \boldsymbol{W} \right\|_F^2, \quad \text{s.t.} \quad \|\boldsymbol{W}\|_0 \le k. \tag{8}$$

We introduce an auxiliary variable $\boldsymbol{D} = \boldsymbol{W}$ and a scaling factor of $\frac{1}{2}$ to simplify the resulting expressions:

$$\min_{\boldsymbol{W}, \boldsymbol{D}} \frac{1}{2} \left( \left\| \widetilde{\boldsymbol{X}} \widehat{\boldsymbol{W}} - \widetilde{\boldsymbol{X}} \boldsymbol{W} \right\|_F^2 + \gamma \left\| \widehat{\boldsymbol{W}} - \boldsymbol{W} \right\|_F^2 \right) + I_{\|\boldsymbol{D}\|_0 \le k}(\boldsymbol{D}), \quad \text{s.t.} \quad \boldsymbol{W} = \boldsymbol{D}, \tag{9}$$

where $I_{\|\boldsymbol{D}\|_0 \le k}(\boldsymbol{D})$ is an indicator function that is 0 if $\|\boldsymbol{D}\|_0 \le k$ and $\infty$ otherwise. The augmented Lagrangian is:

$$L_\rho(\boldsymbol{W}, \boldsymbol{D}, \boldsymbol{V}) = \frac{1}{2} \left( \left\| \widetilde{\boldsymbol{X}} \widehat{\boldsymbol{W}} - \widetilde{\boldsymbol{X}} \boldsymbol{W} \right\|_F^2 + \gamma \left\| \widehat{\boldsymbol{W}} - \boldsymbol{W} \right\|_F^2 \right) + I_{\|\boldsymbol{D}\|_0 \le k}(\boldsymbol{D}) + \langle \boldsymbol{V}, \boldsymbol{W} - \boldsymbol{D} \rangle + \frac{\rho}{2} \|\boldsymbol{W} - \boldsymbol{D}\|_F^2. \tag{10}$$

The ADMM iterations consist of the following updates:

**1. $\boldsymbol{W}$-update:** $\boldsymbol{W}^{(t+1)} = \arg\min_{\boldsymbol{W}} L_\rho(\boldsymbol{W}, \boldsymbol{D}^{(t)}, \boldsymbol{V}^{(t)})$. This involves minimizing:

$$\frac{1}{2} \left( \left\| \widetilde{\boldsymbol{X}} \widehat{\boldsymbol{W}} - \widetilde{\boldsymbol{X}} \boldsymbol{W} \right\|_F^2 + \gamma \left\| \widehat{\boldsymbol{W}} - \boldsymbol{W} \right\|_F^2 \right) + \langle \boldsymbol{V}^{(t)}, \boldsymbol{W} \rangle + \frac{\rho}{2} \left\| \boldsymbol{W} - \boldsymbol{D}^{(t)} \right\|_F^2. \tag{11}$$

Taking the derivative with respect to $\boldsymbol{W}$ and setting it to zero:

$$\nabla_{\boldsymbol{W}} \left[ \frac{1}{2} \left\| \widetilde{\boldsymbol{X}} \widehat{\boldsymbol{W}} - \widetilde{\boldsymbol{X}} \boldsymbol{W} \right\|_F^2 + \frac{\gamma}{2} \left\| \widehat{\boldsymbol{W}} - \boldsymbol{W} \right\|_F^2 + \langle \boldsymbol{V}^{(t)}, \boldsymbol{W} \rangle_F + \frac{\rho}{2} \left\| \boldsymbol{W} - \boldsymbol{D}^{(t)} \right\|_F^2 \right] = 0$$

$$-\widetilde{\boldsymbol{X}}^\top (\widetilde{\boldsymbol{X}} \widehat{\boldsymbol{W}} - \widetilde{\boldsymbol{X}} \boldsymbol{W}) - \gamma(\widehat{\boldsymbol{W}} - \boldsymbol{W}) + \boldsymbol{V}^{(t)} + \rho(\boldsymbol{W} - \boldsymbol{D}^{(t)}) = 0$$

$$(\widetilde{\boldsymbol{X}}^\top \widetilde{\boldsymbol{X}} + \gamma \boldsymbol{I} + \rho \boldsymbol{I}) \boldsymbol{W} = (\widetilde{\boldsymbol{X}}^\top \widetilde{\boldsymbol{X}} + \gamma \boldsymbol{I}) \widehat{\boldsymbol{W}} + \rho \boldsymbol{D}^{(t)} - \boldsymbol{V}^{(t)}.$$

Using the definition $\boldsymbol{H} = (\widetilde{\boldsymbol{X}})^\top \widetilde{\boldsymbol{X}} + \gamma \boldsymbol{I}$:

$$(\boldsymbol{H} + \rho \boldsymbol{I}) \boldsymbol{W}^{(t+1)} = \boldsymbol{H} \widehat{\boldsymbol{W}} + \rho \boldsymbol{D}^{(t)} - \boldsymbol{V}^{(t)}. \tag{12}$$

Thus, the update for $\boldsymbol{W}$ is:

$$\boldsymbol{W}^{(t+1)} = (\boldsymbol{H} + \rho \boldsymbol{I})^{-1} (\boldsymbol{H} \widehat{\boldsymbol{W}} - \boldsymbol{V}^{(t)} + \rho \boldsymbol{D}^{(t)}). \tag{13}$$

**2. $\boldsymbol{D}$-update:** $\boldsymbol{D}^{(t+1)} = \arg\min_{\boldsymbol{D}} L_\rho(\boldsymbol{W}^{(t+1)}, \boldsymbol{D}, \boldsymbol{V}^{(t)})$. This involves minimizing:

$$I_{\|\boldsymbol{D}\|_0 \le k}(\boldsymbol{D}) + \langle \boldsymbol{V}^{(t)}, -\boldsymbol{D} \rangle + \frac{\rho}{2} \left\| \boldsymbol{W}^{(t+1)} - \boldsymbol{D} \right\|_F^2. \tag{14}$$

This can be rewritten by completing the square for terms involving $\boldsymbol{D}$:

$$I_{\|\boldsymbol{D}\|_0 \le k}(\boldsymbol{D}) + \frac{\rho}{2} \left\| \boldsymbol{D} - (\boldsymbol{W}^{(t+1)} + \boldsymbol{V}^{(t)}/\rho) \right\|_F^2 + \text{const.} \tag{15}$$

The solution is obtained by projecting $\boldsymbol{W}^{(t+1)} + \boldsymbol{V}^{(t)}/\rho$ onto the set of matrices with at most $k$ non-zero elements. Let $\widetilde{\boldsymbol{W}}^{(t+1)} = \boldsymbol{W}^{(t+1)} + \boldsymbol{V}^{(t)}/\rho$.

$$\boldsymbol{D}^{(t+1)} = P_k(\widetilde{\boldsymbol{W}}^{(t+1)}), \tag{16}$$

where $P_k(\mathbf{A})$ is an operator that keeps the $k$ elements of $\mathbf{A}$ with the largest magnitudes and sets others to zero. The primary support is $\mathcal{S}^{(t+1)} = \text{Supp}(\boldsymbol{D}^{(t+1)})$.

**3. $\boldsymbol{V}$-update:** The dual variable $\boldsymbol{V}$ is updated as:

$$\boldsymbol{V}^{(t+1)} = \boldsymbol{V}^{(t)} + \rho(\boldsymbol{W}^{(t+1)} - \boldsymbol{D}^{(t+1)}). \tag{17}$$

## A.2 ADMM for Outer Problem

The outer problem (3) aims to find the core weight matrix $\boldsymbol{W}_c$. The problem is:

$$\min_{\boldsymbol{W}_c} \left\| \widetilde{\boldsymbol{X}}\widehat{\boldsymbol{W}} - \widetilde{\boldsymbol{X}}\boldsymbol{W}_c \right\|_F^2 + \gamma \left\| \widehat{\boldsymbol{W}} - \boldsymbol{W}_c \right\|_F^2, \quad \text{s.t.} \quad \|\boldsymbol{W}_c\|_0 \le \alpha k, \text{Supp}(\boldsymbol{W}_c) \subset \mathcal{S}^{(t+1)}, \quad (18)$$

where $\mathcal{S}^{(t+1)} = \text{Supp}(\boldsymbol{D}^{(t+1)})$ is the primary support identified at iteration $t$ from the inner problem's ADMM step.

Similarly, we introduce an auxiliary variable $\boldsymbol{D}_c = \boldsymbol{W}_c$ and a scaling factor of $\frac{1}{2}$ to simplify the resulting expressions:

$$\min_{\boldsymbol{W}_c, \boldsymbol{D}_c} \frac{1}{2} \left( \left\| \widetilde{\boldsymbol{X}}\widehat{\boldsymbol{W}} - \widetilde{\boldsymbol{X}}\boldsymbol{W}_c \right\|_F^2 + \gamma \left\| \widehat{\boldsymbol{W}} - \boldsymbol{W}_c \right\|_F^2 \right) + I_{\|\boldsymbol{D}_c\|_0 \le \alpha k, \text{Supp}(\boldsymbol{D}_c) \subset \mathcal{S}^{(t+1)}}(\boldsymbol{D}_c), \quad \text{s.t.} \quad \boldsymbol{W}_c = \boldsymbol{D}_c. \tag{19}$$

The augmented Lagrangian is:

$$L_{\rho,c}(\boldsymbol{W}_c, \boldsymbol{D}_c, \boldsymbol{V}_c) = \frac{1}{2} \left( \left\| \widetilde{\boldsymbol{X}}\widehat{\boldsymbol{W}} - \widetilde{\boldsymbol{X}}\boldsymbol{W}_c \right\|_F^2 + \gamma \left\| \widehat{\boldsymbol{W}} - \boldsymbol{W}_c \right\|_F^2 \right)$$

$$+ I_{\|\boldsymbol{D}_c\|_0 \le \alpha k, \text{Supp}(\boldsymbol{D}_c) \subset \mathcal{S}^{(t+1)}}(\boldsymbol{D}_c) + \langle \boldsymbol{V}_c, (\boldsymbol{W}_c - \boldsymbol{D}_c) \rangle + \frac{\rho}{2} \|\boldsymbol{W}_c - \boldsymbol{D}_c\|_F^2. \tag{20}$$

**1. $\boldsymbol{W}_c$-update:** $\boldsymbol{W}_c^{(t+1)} = \arg\min_{\boldsymbol{W}_c} L_{\rho,c}(\boldsymbol{W}_c, \boldsymbol{D}_c^{(t)}, \boldsymbol{V}_c^{(t)})$. The objective function for $\boldsymbol{W}_c$ is identical in form to that for $\boldsymbol{W}$. Thus, the derivation is analogous:

$$(\boldsymbol{H} + \rho\boldsymbol{I})\boldsymbol{W}_c^{(t+1)} = \boldsymbol{H}\widehat{\boldsymbol{W}} + \rho\boldsymbol{D}_c^{(t)} - \boldsymbol{V}_c^{(t)}. \tag{21}$$

The update for $\boldsymbol{W}_c$ is:

$$\boldsymbol{W}_c^{(t+1)} = (\boldsymbol{H} + \rho\boldsymbol{I})^{-1}(\boldsymbol{H}\widehat{\boldsymbol{W}} - \boldsymbol{V}_c^{(t)} + \rho\boldsymbol{D}_c^{(t)}). \tag{22}$$

**2. $\boldsymbol{D}_c$-update:** $\boldsymbol{D}_c^{(t+1)} = \arg\min_{\boldsymbol{D}_c} L_{\rho,c}(\boldsymbol{W}_c^{(t+1)}, \boldsymbol{D}_c, \boldsymbol{V}_c^{(t)})$. This involves minimizing:

$$I_{\|\boldsymbol{D}_c\|_0 \le \alpha k, \text{Supp}(\boldsymbol{D}_c) \subset \mathcal{S}^{(t+1)}}(\boldsymbol{D}_c) + \frac{\rho}{2} \left\| \boldsymbol{D}_c - (\boldsymbol{W}_c^{(t+1)} + \boldsymbol{V}_c^{(t)}/\rho) \right\|_F^2. \tag{23}$$

Let $\widetilde{\boldsymbol{W}}_c^{(t+1)} = \boldsymbol{W}_c^{(t+1)} + \boldsymbol{V}_c^{(t)}/\rho$. The solution is obtained by first ensuring the support constraint $\text{Supp}(\boldsymbol{D}_c) \subset \mathcal{S}^{(t+1)}$ and then projecting onto the set of matrices with at most $\alpha k$ non-zero elements. This is achieved by masking $\widetilde{\boldsymbol{W}}_c^{(t+1)}$ with the primary support $\mathcal{S}^{(t+1)}$ and then selecting the $\alpha k$ largest magnitude elements from this masked matrix:

$$\boldsymbol{D}_c^{(t+1)} = P_{\alpha k}(\widetilde{\boldsymbol{W}}_c^{(t+1)} \odot \mathcal{S}^{(t+1)}). \tag{24}$$

**3. $\boldsymbol{V}_c$-update:** The dual variable $\boldsymbol{V}_c$ is updated as:

$$\boldsymbol{V}_c^{(t+1)} = \boldsymbol{V}_c^{(t)} + \rho(\boldsymbol{W}_c^{(t+1)} - \boldsymbol{D}_c^{(t+1)}). \tag{25}$$

These derivations provide the update rules for $\boldsymbol{W}, \boldsymbol{D}, \boldsymbol{V}$ and $\boldsymbol{W}_c, \boldsymbol{D}_c, \boldsymbol{V}_c$ as presented in Section 3.2.1 of the main paper.

# B Proofs of Theorem 1

*Proof.* Since we do not modify the update steps for $\boldsymbol{W}$ and $\boldsymbol{D}$ compared to ALPS [30], the convergence proof of ALPS remains valid for $\boldsymbol{W}$ and $\boldsymbol{D}$. Here, we further extend the proof in [30] to analyze the convergence of $\boldsymbol{W}_c$ and $\boldsymbol{D}_c$.

For the sake of conciseness, throughout the proof, we denote $\mathbf{H} = \mathbf{X}^\top\mathbf{X}$ and $\mathbf{G} = \mathbf{X}^\top\mathbf{X}\widehat{\boldsymbol{W}}$. To establish the theorem, we first present the following two lemmas. The proofs of these two lemmas are given in Section B.1 and B.2, respectively.

**Lemma 1.** *Let* $\left\{\mathbf{D}_c^{(t)}\right\}_{t=0}^{\infty}$ *and* $\left\{\mathbf{V}_c^{(t)}\right\}_{t=0}^{\infty}$ *be the sequence generated by MOSP. Then for any* $t \geq 0$, *it holds*

$$\|\mathbf{V}_c^{(t+1)}\|_F \leq \|\mathbf{G} - \mathbf{H}\mathbf{D}_c^{(t)}\|_F + \frac{\|\mathbf{H}\mathbf{V}_c^{(t)}\|_F}{\rho_t} \tag{26}$$

*and*

$$\|\mathbf{D}_c^{(t+1)} - \mathbf{D}_c^{(t)}\|_F \leq \frac{2}{\rho_t}\left(\|\mathbf{G} - \mathbf{H}\mathbf{D}_c^{(t)}\|_F + \frac{\|\mathbf{H}\mathbf{V}_c^{(t)}\|_F}{\rho_t}\right). \tag{27}$$

**Lemma 2.** *Let* $\left\{\mathbf{D}_c^{(t)}\right\}_{t=0}^{\infty}$, $\left\{\mathbf{W}_c^{(t)}\right\}_{t=0}^{\infty}$ *and* $\left\{\mathbf{V}_c^{(t)}\right\}_{t=0}^{\infty}$ *be the sequence generated by MOSP. Suppose* $\{\rho_t\}_{t=0}^{\infty}$ *is non-decreasing. Then for any* $t \geq 0$, *it holds*

$$\|\mathbf{D}_c^{(t)}\|_F + \frac{\|\mathbf{V}_c^{(t)}\|_F}{\rho_t} \leq \left[\prod_{s=0}^{t-1}\left(1 + \frac{3\|\mathbf{H}\|_2}{\rho_s}\right)\right] \cdot \left(\|\mathbf{D}_c^{(0)}\|_F + \frac{\|\mathbf{V}_c^{(0)}\|_F}{\rho_0} + \sum_{s=0}^{t-1}\frac{3\|\mathbf{G}\|_F}{\rho_s}\right) \tag{28}$$

Returning to the proof of the main theorem, combining Lemma 2 with the initialization of our method MOSP gives

$$
\begin{aligned}
\|\mathbf{D}_c^{(t)}\|_F + \frac{\|\mathbf{V}_c^{(t)}\|_F}{\rho_t} &\leq \left[\prod_{s=0}^{t-1}\left(1 + \frac{3\|\mathbf{H}\|_2}{\rho_s}\right)\right] \cdot \left(\|\mathbf{D}_c^{(0)}\|_F + \frac{\|\mathbf{V}_c^{(0)}\|_F}{\rho_0} + \sum_{s=0}^{t-1}\frac{3\|\mathbf{G}\|_F}{\rho_s}\right) \\
&\leq \exp\left(3\|\mathbf{H}\|_2\sum_{s=0}^{\infty}\frac{1}{\rho_s}\right) \cdot \left(\|\mathbf{G}\|_F + 3\|\mathbf{G}\|_F\sum_{s=0}^{\infty}\frac{1}{\rho_s}\right)
\end{aligned}
\tag{29}
$$

Let

$$C(\mathbf{X}, \widehat{\mathbf{W}}, \rho_0, t_u, \hat{\tau}) := 2\|\mathbf{G}\|_F + 2\|\mathbf{H}\|_2\left(\exp\left(3\|\mathbf{H}\|_2\sum_{s=0}^{\infty}\frac{1}{\rho_s}\right) \cdot \left(\|\mathbf{G}\|_F + 3\|\mathbf{G}\|_F\sum_{s=0}^{\infty}\frac{1}{\rho_s}\right)\right) \tag{30}$$

be the constant depending on $\mathbf{X}$, $\widehat{\mathbf{W}}$ and $\sum_{s=0}^{\infty}1/\rho_s$. Lemma 1 together with (29) leads to

$$
\begin{aligned}
\|\mathbf{V}_c^{(t+1)}\|_F &\leq \|\mathbf{G} - \mathbf{H}\mathbf{D}_c^{(t)}\|_F + \frac{\|\mathbf{H}\mathbf{V}_c^{(t)}\|_F}{\rho_t} \\
&\leq \|\mathbf{G}\|_F + \|\mathbf{H}\|_2\left(\|\mathbf{D}_c^{(t)}\|_F + \frac{\|\mathbf{V}_c^{(t)}\|_F}{\rho_t}\right) \leq \frac{1}{2}C(\mathbf{X}, \widehat{\mathbf{W}}, \rho_0, t_u, \hat{\tau})
\end{aligned}
\tag{31}
$$

and

$$\|\mathbf{D}_c^{(t+1)} - \mathbf{D}_c^{(t)}\|_F \leq \frac{2}{\rho_t}\left(\|\mathbf{G} - \mathbf{H}\mathbf{D}_c^{(t)}\|_F + \frac{\|\mathbf{H}\mathbf{V}_c^{(t)}\|_F}{\rho_t}\right) \leq \frac{C(\mathbf{X}, \widehat{\mathbf{W}}, \rho_0, t_u, \hat{\tau})}{\rho_t}. \tag{32}$$

It then follows from $\mathbf{W}_c^{(t+1)} - \mathbf{D}_c^{(t+1)} = (\mathbf{V}_c^{(t+1)} - \mathbf{V}_c^{(t)})/\rho_t$ that

$$\|\mathbf{W}_c^{(t+1)} - \mathbf{D}_c^{(t+1)}\|_F \leq \frac{\|\mathbf{V}_c^{(t+1)}\|_F + \|\mathbf{V}_c^{(t)}\|_F}{\rho_t} \leq \frac{C(\mathbf{X}, \widehat{\mathbf{W}}, \rho_0, t_u, \hat{\tau})}{\rho_t}. \tag{33}$$

Therefore, we prove the desired inequality. Since $\sum_{s=0}^{\infty}1/\rho_s < \infty$, $\{\mathbf{D}_c\}_{t=0}^{\infty}$ is a Cauchy sequence, and therefore there exists a matrix $\bar{\mathbf{D}}_c$ such that $\mathbf{D}_c^{(t)} \to \bar{\mathbf{D}}_c$. It follows from $\|\mathbf{W}_c^{(t+1)} - \mathbf{D}_c^{(t+1)}\|_F \to 0$ that $\mathbf{W}_c^{(t)} \to \bar{\mathbf{D}}_c$. The proof is completed.

$\square$

## B.1 Proof of Lemma 1

*Proof.* According to the $\mathbf{W}-$update rule in (5), it holds

$$
\begin{aligned}
\mathbf{W}_c^{(t+1)} - \mathbf{D}_c^{(t)} + \frac{\mathbf{V}_c^{(t)}}{\rho^{(t)}} &= (\mathbf{H} + \rho_t \mathbf{I})^{-1}(\mathbf{G} - \mathbf{V}_c^{(t)} + \rho_t \mathbf{D}_c^{(t)}) - \mathbf{D}_c^{(t)} + \frac{\mathbf{V}_c^{(t)}}{\rho^{(t)}} \\
&= \left((\mathbf{H} + \rho_t \mathbf{I})^{-1}\rho_t - \mathbf{I}\right)\mathbf{D}_c^{(t)} + (\mathbf{H} + \rho_t \mathbf{I})^{-1}(\mathbf{G} - \mathbf{V}_c^{(t)}) + \frac{\mathbf{V}_c^{(t)}}{\rho_t} \\
&= -\frac{1}{\rho_t}\left(\mathbf{I} + \frac{\mathbf{H}}{\rho_t}\right)^{-1}\mathbf{H}\mathbf{D}_c^{(t)} + \frac{1}{\rho_t}\left(\mathbf{I} + \frac{\mathbf{H}}{\rho_t}\right)^{-1}(\mathbf{G} - \mathbf{V}_c^{(t)}) + \frac{\mathbf{V}_c^{(t)}}{\rho_t} \\
&= \frac{1}{\rho_t}\left(\mathbf{I} + \frac{\mathbf{H}}{\rho_t}\right)^{-1}(\mathbf{G} - \mathbf{H}\mathbf{D}_c^{(t)}) + \frac{1}{\rho_t}\left[\mathbf{I} - \left(\mathbf{I} + \frac{\mathbf{H}}{\rho_t}\right)^{-1}\right]\mathbf{V}_c^{(t)} \\
&= \frac{1}{\rho_t}\left(\mathbf{I} + \frac{\mathbf{H}}{\rho_t}\right)^{-1}\left(\mathbf{G} - \mathbf{H}\mathbf{D}_c^{(t)} + \frac{\mathbf{H}\mathbf{V}_c^{(t)}}{\rho_t}\right)
\end{aligned}
\tag{34}
$$

Therefore, we obtain

$$
\begin{aligned}
\left\|\mathbf{W}_c^{(t+1)} - \mathbf{D}_c^{(t)} + \frac{\mathbf{V}_c^{(t)}}{\rho^{(t)}}\right\|_F &\leq \frac{1}{\rho_t}\left\|\left(\mathbf{I} + \frac{\mathbf{H}}{\rho_t}\right)^{-1}\right\|_2 \left\|\mathbf{G} - \mathbf{H}\mathbf{D}_c^{(t)} + \frac{\mathbf{H}\mathbf{V}_c^{(t)}}{\rho_t}\right\|_F \\
&\leq \frac{1}{\rho_t}\left\|\mathbf{G} - \mathbf{H}\mathbf{D}_c^{(t)} + \frac{\mathbf{H}\mathbf{V}_c^{(t)}}{\rho_t}\right\|_F \\
&\leq \frac{1}{\rho_t}\left(\|\mathbf{G} - \mathbf{H}\mathbf{D}_c^{(t)}\|_F + \frac{\|\mathbf{H}\mathbf{V}_c^{(t)}\|}{\rho_t}\right).
\end{aligned}
\tag{35}
$$

Denote $\widetilde{\mathcal{I}}^{(t)} := \{(i,j) \in [c] \times [d] \mid \mathbf{D}_{ij}^{(t)} = 0\}$, $\widetilde{\mathcal{I}}_c^{(t)} := \{(i,j) \in [c] \times [d] \mid (\mathbf{D}_c^{(t)})_{ij} = 0\}$. It follows from the $\mathbf{D}-$update rule and the definition of the projection operator that

$$
\begin{aligned}
\left\|\mathbf{D}_c^{(t+1)} - \mathbf{W}_c^{(t+1)} - \frac{\mathbf{V}_c^{(t)}}{\rho_t}\right\|_F^2 &= \min_{\substack{\mathcal{I} \subseteq \text{Supp}(\boldsymbol{D}^{(t+1)}) \\ |\mathcal{I}| = (1-l)k}} \sum_{(i,j) \in \mathcal{I}}\left(\mathbf{W}_c^{(t+1)} + \frac{\mathbf{V}_c^{(t)}}{\rho_t}\right)_{i,j}^2 \\
&+ \sum_{(i,j) \in \widetilde{\mathcal{I}}^{(t+1)}}\left(\mathbf{W}_c^{(t+1)} + \frac{\mathbf{V}_c^{(t)}}{\rho_t}\right)_{i,j}^2
\end{aligned}
\tag{36}
$$

By definition, note that $\text{Supp}(\boldsymbol{D}^{(t+1)}) \cup \widetilde{\mathcal{I}}^{(t+1)} = [c] \times [d]$ and $\text{Supp}(\boldsymbol{D}^{(t+1)}) \cap \widetilde{\mathcal{I}}^{(t+1)} = \phi$, then we can write $\widetilde{\mathcal{I}}_c^{(t)} = \left(\widetilde{\mathcal{I}}_c^{(t)} \cap \text{Supp}(\boldsymbol{D}^{(t+1)})\right) \cup \left(\widetilde{\mathcal{I}}_c^{(t)} \cap \widetilde{\mathcal{I}}^{(t+1)}\right)$. Then from $|\widetilde{\mathcal{I}}^{(t+1)}| = cd - k$, we must have $|\widetilde{\mathcal{I}}_c^{(t)} \cap \widetilde{\mathcal{I}}^{(t+1)}| \leq cd - k$. And since $|\widetilde{\mathcal{I}}_c^{(t)}| = cd - lk$ and $\widetilde{\mathcal{I}}_c^{(t)} = \left(\widetilde{\mathcal{I}}_c^{(t)} \cap \text{Supp}(\boldsymbol{D}^{(t+1)})\right) \cup \left(\widetilde{\mathcal{I}}_c^{(t)} \cap \widetilde{\mathcal{I}}^{(t+1)}\right)$, we should have $|\widetilde{\mathcal{I}}_c^{(t)} \cap \text{Supp}(\boldsymbol{D}^{(t+1)})| \geq (1 - l)k$. Therefore, assume $|\widetilde{\mathcal{I}}_c^{(t)} \cap \text{Supp}(\boldsymbol{D}^{(t+1)})| = (1-l)k + \Delta$ with $\Delta \geq 0$, which also directly gives us $|\widetilde{\mathcal{I}}_c^{(t)} \cap \widetilde{\mathcal{I}}^{(t+1)}| \leq cd - k - \Delta$ we first should have:

$$
\begin{aligned}
\min_{\substack{\mathcal{I} \subseteq \text{Supp}(\boldsymbol{D}^{(t+1)}) \\ |\mathcal{I}| = (1-l)k}} \sum_{(i,j) \in \mathcal{I}}\left(\mathbf{W}_c^{(t+1)} + \frac{\mathbf{V}_c^{(t)}}{\rho_t}\right)_{i,j}^2 &\leq \min_{\substack{\mathcal{I} \subseteq \left(\widetilde{\mathcal{I}}_c^{(t)} \cap \text{Supp}(\boldsymbol{D}^{(t+1)})\right) \\ |\mathcal{I}| = (1-l)k}} \sum_{(i,j) \in \mathcal{I}}\left(\mathbf{W}_c^{(t+1)} + \frac{\mathbf{V}_c^{(t)}}{\rho_t}\right)_{i,j}^2 \\
&= \min_{\substack{\mathcal{I} \subseteq \left(\widetilde{\mathcal{I}}_c^{(t)} \cap \text{Supp}(\boldsymbol{D}^{(t+1)})\right) \\ |\mathcal{I}| = (1-l)k}} \sum_{(i,j) \in \mathcal{I}}\left(\mathbf{W}_c^{(t+1)} - \mathbf{D}_c^{(t)} + \frac{\mathbf{V}_c^{(t)}}{\rho_t}\right)_{i,j}^2
\end{aligned}
\tag{37}
$$

Regarding the second term in (36), we have:

$$\sum_{(i,j)\in\widetilde{\mathcal{I}}^{(t+1)}}\left(\mathbf{W}_c^{(t+1)}+\frac{\mathbf{V}_c^{(t)}}{\rho_t}\right)_{i,j}^2 = \sum_{(i,j)\in\widetilde{\mathcal{I}}_c^{(t)}\cap\widetilde{\mathcal{I}}^{(t+1)}}\left(\mathbf{W}_c^{(t+1)}+\frac{\mathbf{V}_c^{(t)}}{\rho_t}\right)_{i,j}^2$$

$$+ \sum_{(i,j)\in\widetilde{\mathcal{I}}^{(t+1)}/\widetilde{\mathcal{I}}_c^{(t)}}\left(\mathbf{W}_c^{(t+1)}+\frac{\mathbf{V}_c^{(t)}}{\rho_t}\right)_{i,j}^2$$

By definition, we should have $|\widetilde{\mathcal{I}}^{(t+1)}/\widetilde{\mathcal{I}}_c^{(t)}| = \Delta$, which also matches the number of additional elements in $|\widetilde{\mathcal{I}}_c^{(t)} \cap \mathtt{Supp}(\boldsymbol{D}^{(t+1)})|$ other than the smallest $(1-l)k$ elements in it. Then since all elements in $\widetilde{\mathcal{I}}^{(t+1)}$ corresponds to the smallest $cd-k$ elements in $\mathbf{W}^{(t+1)}+\frac{\mathbf{V}^{(t)}}{\rho_t}$, we should have:

$$\max_{\substack{\mathcal{I}\subseteq(\widetilde{\mathcal{I}}_c^{(t)}\cap\mathtt{Supp}(\boldsymbol{D}^{(t+1)}))\\|\mathcal{I}|=\Delta}}\sum_{(i,j)\in\mathcal{I}}\left(\mathbf{W}_c^{(t+1)}+\frac{\mathbf{V}_c^{(t)}}{\rho_t}\right)_{i,j}^2 \geq \sum_{(i,j)\in\widetilde{\mathcal{I}}^{(t+1)}/\widetilde{\mathcal{I}}_c^{(t)}}\left(\mathbf{W}_c^{(t+1)}+\frac{\mathbf{V}_c^{(t)}}{\rho_t}\right)_{i,j}^2$$

Thus we have:

$$\sum_{(i,j)\in\widetilde{\mathcal{I}}^{(t+1)}}\left(\mathbf{W}_c^{(t+1)}+\frac{\mathbf{V}_c^{(t)}}{\rho_t}\right)_{i,j}^2 \leq \sum_{(i,j)\in\widetilde{\mathcal{I}}_c^{(t)}\cap\widetilde{\mathcal{I}}^{(t+1)}}\left(\mathbf{W}_c^{(t+1)}+\frac{\mathbf{V}_c^{(t)}}{\rho_t}\right)_{i,j}^2$$

$$+ \max_{\substack{\mathcal{I}\subseteq(\widetilde{\mathcal{I}}_c^{(t)}\cap\mathtt{Supp}(\boldsymbol{D}^{(t+1)}))\\|\mathcal{I}|=\Delta}}\sum_{(i,j)\in\mathcal{I}}\left(\mathbf{W}_c^{(t+1)}+\frac{\mathbf{V}_c^{(t)}}{\rho_t}\right)_{i,j}^2 \qquad (38)$$

Then combining (37) and (38) will give us:

$$\left\|\mathbf{D}_c^{(t+1)}-\mathbf{W}_c^{(t+1)}-\frac{\mathbf{V}_c^{(t)}}{\rho_t}\right\|_F^2 = \min_{\substack{\mathcal{I}\subseteq\mathtt{Supp}(\boldsymbol{D}^{(t+1)})\\|\mathcal{I}|=(1-l)k}}\sum_{(i,j)\in\mathcal{I}}\left(\mathbf{W}_c^{(t+1)}+\frac{\mathbf{V}_c^{(t)}}{\rho_t}\right)_{i,j}^2$$

$$+ \sum_{(i,j)\in\widetilde{\mathcal{I}}^{(t+1)}}\left(\mathbf{W}_c^{(t+1)}+\frac{\mathbf{V}_c^{(t)}}{\rho_t}\right)_{i,j}^2$$

$$\leq \sum_{(i,j)\in(\widetilde{\mathcal{I}}_c^{(t)}\cap\mathtt{Supp}(\boldsymbol{D}^{(t+1)}))}\left(\mathbf{W}_c^{(t+1)}+\frac{\mathbf{V}_c^{(t)}}{\rho_t}\right)_{i,j}^2 + \sum_{(i,j)\in\widetilde{\mathcal{I}}_c^{(t)}\cap\widetilde{\mathcal{I}}^{(t+1)}}\left(\mathbf{W}_c^{(t+1)}+\frac{\mathbf{V}_c^{(t)}}{\rho_t}\right)_{i,j}^2 \qquad (39)$$

$$\leq \sum_{(i,j)\in\widetilde{\mathcal{I}}_c^{(t)}}\left(\mathbf{W}_c^{(t+1)}-\mathbf{D}_c^{(t)}+\frac{\mathbf{V}_c^{(t)}}{\rho_t}\right)_{i,j}^2$$

$$\leq \left\|\mathbf{W}_c^{(t+1)}-\mathbf{D}_c^{(t)}+\frac{\mathbf{V}_c^{(t)}}{\rho_t}\right\|_F^2$$

Together with (35), we get

$$\left\|\mathbf{D}_c^{(t+1)}-\mathbf{W}_c^{(t+1)}-\frac{\mathbf{V}_c^{(t)}}{\rho_t}\right\|_F \leq \frac{1}{\rho_t}\left(\|\mathbf{G}-\mathbf{H}\mathbf{D}_c^{(t)}\|_F+\frac{\|\mathbf{H}\mathbf{V}_c^{(t)}\|}{\rho_t}\right). \qquad (40)$$

It then follows from the $\mathbf{V}-$update rule that

$$\frac{\|\mathbf{V}_c^{(t+1)}\|_F}{\rho_t} = \left\|\mathbf{D}_c^{(t+1)}-\mathbf{W}_c^{(t+1)}-\frac{\mathbf{V}_c^{(t)}}{\rho_t}\right\|_F \leq \frac{1}{\rho_t}\left(\|\mathbf{G}-\mathbf{H}\mathbf{D}_c^{(t)}\|_F+\frac{\|\mathbf{H}\mathbf{V}_c^{(t)}\|}{\rho_t}\right) \qquad (41)$$

This establishes the inequality (26). Furthermore, by summing up (35) and (40) and applying the triangle inequality, we verify the inequality (27). □

### B.2 Proof of Lemma 2

*Proof.* It follows from Lemma 1 that

$$\|\mathbf{V}_c^{(t+1)}\|_F \le \|\mathbf{G} - \mathbf{H}\mathbf{D}_c^{(t)}\|_F + \frac{\|\mathbf{H}\mathbf{V}_c^{(t)}\|_F}{\rho_t}$$

$$\le \|\mathbf{H}\|_2\|\mathbf{D}_c^{(t)}\|_F + \|\mathbf{G}\|_F + \frac{\|\mathbf{H}\|_2\|\mathbf{V}_c^{(t)}\|_F}{\rho_t} \tag{42}$$

and

$$\|\mathbf{D}_c^{(t+1)} - \mathbf{D}_c^{(t)}\|_F \le \frac{2}{\rho_t}\left(\|\mathbf{G} - \mathbf{H}\mathbf{D}_c^{(t)}\|_F + \frac{\|\mathbf{H}\mathbf{V}_c^{(t)}\|_F}{\rho_t}\right)$$

$$\le \frac{2}{\rho_t}\left(\|\mathbf{H}\|_2\|\mathbf{D}_c^{(t)}\|_F + \|\mathbf{G}\|_F + \frac{\|\mathbf{H}\|_2\|\mathbf{V}_c^{(t)}\|_F}{\rho_t}\right). \tag{43}$$

This further implies

$$\|\mathbf{D}_c^{(t+1)}\|_F \le \left(1 + \frac{2\|\mathbf{H}\|_2}{\rho_t}\right)\|\mathbf{D}_c^{(t)}\|_F + \frac{2\|\mathbf{G}\|_F}{\rho_t} + \frac{2\|\mathbf{H}\|_2\|\mathbf{V}_c^{(t)}\|_F}{\rho_t^2} \tag{44}$$

Combining inequalities (42) and (44) yields

$$\|\mathbf{D}_c^{(t+1)}\|_F + \frac{\|\mathbf{V}_c^{(t+1)}\|_F}{\rho_{t+1}} \le \|\mathbf{D}_c^{(t+1)}\|_F + \frac{\|\mathbf{V}_c^{(t+1)}\|_F}{\rho_t}$$

$$\le \left(1 + \frac{3\|\mathbf{H}\|_2}{\rho_t}\right)\|\mathbf{D}_c^{(t)}\|_F + \frac{3\|\mathbf{G}\|_F}{\rho_t} + \frac{3\|\mathbf{H}\|_2\|\mathbf{V}_c^{(t)}\|_F}{\rho_t^2} \tag{45}$$

$$\le \left(1 + \frac{3\|\mathbf{H}\|_2}{\rho_t}\right)\left(\|\mathbf{D}_c^{(t)}\|_F + \frac{\|\mathbf{V}_c^{(t)}\|_F}{\rho_t}\right) + \frac{3\|\mathbf{G}\|_F}{\rho_t}$$

Denote $a_t := \|\mathbf{D}_c^{(t)}\|_F + \|\mathbf{V}_c^{(t)}\|_F/\rho_t$, then the above inequality can be rewritten as

$$a_{t+1} \le \left(1 + \frac{3\|\mathbf{H}\|_2}{\rho_t}\right)a_t + \frac{3\|\mathbf{G}\|_F}{\rho_t} \tag{46}$$

Therefore,

$$\frac{a_{t+1}}{\prod_{s=0}^t(1 + 3\|\mathbf{H}\|_2/\rho_k)} \le \frac{a_t}{\prod_{s=0}^{t-1}(1 + 3\|\mathbf{H}\|_2/\rho_k)} + \frac{3\|\mathbf{G}\|_F}{\rho_t\prod_{s=0}^t(1 + 3\|\mathbf{H}\|_2/\rho_k)}$$

$$\le \frac{a_t}{\prod_{s=0}^{t-1}(1 + 3\|\mathbf{H}\|_2/\rho_k)} + \frac{3\|\mathbf{G}\|_F}{\rho_t} \tag{47}$$

It then follows from telescoping that

$$\frac{a_t}{\prod_{s=0}^{t-1}(1 + 3\|\mathbf{H}\|_2/\rho_k)} \le a_0 + \sum_{s=0}^{t-1}\frac{3\|\mathbf{G}\|_F}{\rho_t} \tag{48}$$

Recalling the definition of $a_t$ completes the proof. $\square$

## C  Details of the Refinement Stage

Algorithm 1 details the refinement stage, which further refines each model using Projected Conjugate Gradient (PCG), following ALPS [30].

## D  Experimental Details

In this section, we provide more details about the experiment.

We perform all experiments on a GPU server equipped with 8 NVIDIA A6000 GPUs, each with 48GB of memory. However, for each individual experiment, we utilize only a single GPU. We use PyTorch version 2.0.0 [35]. The total time of execution for all reproduce all experiments is around 100 hours.

---

**Algorithm 1** PCG with vectorization [30] for $i$th task.

---

**Require:** Support $\mathcal{S}$, pre-conditioner $\mathbf{M} = \mathrm{Diag}(\mathbf{H})$, initial solution $\mathbf{W}_i$
1: Set $\mathbf{R}^{(0)} := \mathbf{H}_i(\widehat{\mathbf{W}} - \mathbf{W}_i)$
2: Project $\mathbf{R}^{(0)}$ onto the support $\mathcal{S}$ by setting all elements outside the support to zero.
3: Set $\mathbf{Z}^{(0)} = \mathbf{M}^{-1}\mathbf{R}^{(0)}$ and $\mathbf{P}^{(0)} = \mathbf{Z}^{(0)}$
4: **for** $t = 0, 1, \ldots$ **do**
5: $\qquad \alpha^{(t)} = \dfrac{\mathrm{sum}(\mathbf{R}^{(t)} \odot \mathbf{Z}^{(t)}, \mathrm{axis} = 0)}{\mathrm{sum}(\mathbf{P}^{(t)} \odot \mathbf{Q}^{(t)}, \mathrm{axis} = 0)}$
6: $\qquad \mathbf{W}_i^{(t+1)} = \mathbf{W}_i^{(t)} + \alpha^{(t)} \odot \mathbf{P}^{(t)}$
7: $\qquad \mathbf{R}^{(t+1)} = \mathbf{R}^{(t)} - \alpha^{(t)} \odot \mathbf{H}_i\mathbf{P}^{(t)}$
8: $\qquad$ Project $\mathbf{R}^{(t+1)}$ onto the support $\mathcal{S}$ by setting all elements outside the support to zero.
9: $\qquad \mathbf{Z}^{(t+1)} = \mathbf{M}^{-1}\mathbf{R}^{(t+1)}$
10: $\qquad$ **if** $\mathbf{R}^{(t+1)}$ is sufficiently small **then**
11: $\qquad\qquad$ **break**
12: $\qquad$ **end if**
13: $\qquad \beta^{(t)} = \dfrac{\mathrm{sum}(\mathbf{R}^{(t+1)} \odot \mathbf{Z}^{(t+1)}, \mathrm{axis} = 0)}{\mathrm{sum}(\mathbf{R}^{(t)} \odot \mathbf{Z}^{(t)}, \mathrm{axis} = 0)}$
14: $\qquad \mathbf{P}^{(t+1)} := \mathbf{Z}^{(t+1)} + \beta^{(t)} \odot \mathbf{P}^{(t)}$
15: **end for**

---

**Preference Vectors.** For MOSP, we sample user preferences ($\boldsymbol{\lambda}$) to trace the Pareto front. In the two-objective scenario, we employ 11 uniform preference vectors (i.e., $[1, 0], [0.9, 0.1], \ldots, [0.1, 0.9], [0, 1]$). For the three-objective scenario, we utilize 36 uniform preference vectors. These vectors are generated by defining a 2-simplex (an equilateral triangle where components sum to 1) and selecting points $(\lambda_1, \lambda_2, \lambda_3)$ such that $\lambda_i = k_i/S$ for $k_i \in \mathbb{N}_0$ and $\sum_i k_i = S$. For 36 vectors, this corresponds to $S = 7$. The meshzoo package [2] is used to help generate the vertices of a triangular mesh, which correspond to these preference vectors. Each preference vector $\boldsymbol{\lambda}$ produces a distinct pruned model on the Pareto front.

**Hypervolume.** The Hypervolume (HV) indicator [47, 21] is a popular metric in multi-objective optimization (MOO), offering a measure of the quality of an obtained solution set by quantifying the volume of the dominated portion of the objective space. Formally, for a given solution set $P$ and a reference point $r \in \mathbb{R}^m$ (where $m$ is the number of objectives), the HV is defined as:

$$\mathrm{HV}(P, r) = \Lambda \left( \bigcup_{p \in P} \{q \in \mathbb{R}^m \mid p \succeq q \succeq r\} \right), \tag{49}$$

where $\Lambda$ denotes the Lebesgue measure, and $p \succeq q$ means $p$ dominates or is equal to $q$ (assuming maximization of objectives). A larger HV is preferable, indicating a better approximation of the true Pareto front.

**Hyperparameter Setting.** As mentioned in the main paper, we set both $\alpha$ and $p$ to 0.5. For other hyperparameters, we follow the settings in [30]. Specifically, we set the initial penalty parameter $\rho_0 = 0.1$. We update $\rho$ every 3 iterations based on a step function that depends on the current value of $\rho_t$ and $s_t := |\mathrm{Supp}(\mathbf{D}^{(t)})\Delta\mathrm{Supp}(\mathbf{D}^{(t-3)})|$. The term $s_t$ represents the number of elements in the symmetric difference between the support of $\mathbf{D}$ at iteration $t$ and iteration $t - 3$. Specifically, the update rule is:

$$\rho_{t+1} = \begin{cases} 1.3\rho_t & \text{if } s_t \geq 0.1k, \\ 1.2\rho_t & \text{if } s_t \geq 0.005k, \\ 1.1\rho_t & \text{if } s_t \geq 1. \end{cases} \tag{50}$$

where $k$ is the total number of elements or parameters being considered for pruning. If $s_t = 0$, it indicates that $\rho$ is sufficiently large and the support has stabilized. For stage 2, the task-specific ADMM, we use a fixed $\rho = 0.5$. We set $\gamma$ to $0.01\mathrm{Tr}(\boldsymbol{X}^\top\boldsymbol{X})$. We refine each model with 10 PCG iterations.

---

[2] https://github.com/meshpro/meshzoo

**Dataset.**  We use three publicly available datasets for our experiments:

- **C4 (Colossal Clean Crawled Corpus)**[36]: *ODC-BY License.* This is a large-scale, cleaned version of the Common Crawl dataset, containing primarily English text. Following common practice [37, 15, 30], we use a subset of C4 dataset. We use the same data split as [30].
- **GSM8K (Grade School Math 8K)**[9]: *MIT License.* A dataset comprising high-quality grade school mathematics word problems designed to evaluate the multi-step reasoning capabilities of models. We follow the official data split.
- **Python Code**[5]: *CC-BY-4.0 License.* An instruction-following dataset tailored for Python code generation, styled after the Alpaca dataset. We follow the official data split.

**Licenses for Models.**  The licenses for the models are as follows: for the LLaMA-2 series, the license is the "LLaMA 2 Community License Agreement"; for the LLaMA-3 series, it is the "LLaMA 3 Community License Agreement"; and for the OPT series, the license is the "OPT License Agreement."

# E  Additional Experimental Results

**Extension to More Objectives.**  In this section, in addition to the three objectives discussed in Section 4.3, we introduce a fourth objective: multilingual performance evaluated on ChineseWebText 2.0. All other experimental settings remain the same as in Section 4.3. The results, summarized in Table 5, show that MOSP continues to effectively identify diverse Pareto-optimal solutions across all four objectives.

Table 5: Test PPL of Llama-2-7B pruned to 70% unstructured sparsity with different preference vectors.

| Method | C4 | Code | GSM8K | Chinese |
|---|---|---|---|---|
| SparseGPT | 30.52 | 3.21 | 3.60 | 16.57 |
| ALPS | 20.57 | 2.67 | 3.17 | 9.53 |
| MOSP($\lambda^{(1)} = [1, 0, 0, 0]$) | **19.17** | 3.37 | 3.42 | 14.31 |
| MOSP($\lambda^{(2)} = [0, 1, 0, 0]$) | 26.69 | **2.57** | 3.50 | 17.31 |
| MOSP($\lambda^{(3)} = [0, 0, 1, 0]$) | 25.07 | 3.25 | **3.07** | 18.30 |
| MOSP($\lambda^{(4)} = [0, 0, 0, 1]$) | 22.52 | 3.26 | 3.51 | **8.17** |
| MOSP($\lambda^{(5)} = [0.07, 0.07, 0.07, 0.8]$) | 20.19 | 2.97 | 3.34 | 8.52 |

**Efficiency Improvement.**  Our observed speedups are consistent with those reported in SparseGPT [15]. This is because speedups are fundamentally determined by model sparsity and hardware; the pruning algorithm used makes only a minimal difference. We refer the reader to SparseGPT [15] for more comprehensive results.

**Additional Visualization Results.**  We present additional visualization results across various models and sparsity levels. Figure 7 compares MOSP and SparseGPT when pruning Llama-2-7B to 50%, 60%, 70%, 80% unstructured sparsity and 2:4 semi-structured sparsity. Figures 8 and 9 show the performance of Llama-2-7B pruned to 2:4 semi-structured sparsity and 80% sparsity, respectively. Figures 10, 11, 12, 13, 14 show the performance of OPT-1.3B, OPT-2.7B, Llama-3-8B, Llama-2-13B, and OPT-30B at 70% sparsity. These results demonstrate that the proposed method consistently provides diverse trade-off solutions.

# F  Further Discussion on the Motivation

Our primary motivation is to serve the diverse range of user preferences that exist between the extremes of "pure generality" and various "pure specializations." For instance, different deployments of a model on edge devices may require different balances between capabilities: One user might prioritize language understanding (60% importance) over coding (30%) and math (10%), while another may require strong mathematical reasoning (80% importance). As the number of objectives

increases, these preference combinations grow exponentially. MOSP efficiently addresses this by obtaining a set of trade-off models in a single pruning run. This allows users to select the best-fit model for their specific needs post-training, avoiding the significant computational cost of re-pruning for every new preference. For instance, handling 100 distinct user preferences with traditional methods would take over 100 hours, whereas MOSP accomplishes this in just 1.36 hours. This approach also uniquely allows for dynamic preference adjustments.

**Comparison with Single-Objective Baselines.** To further contextualize these trade-offs, we compared MOSP against single-objective optimization by running the ALPS baseline on each task individually. The results are shown in Table 6. As can be seen, single-objective ALPS achieves optimal performance on its target task but shows significant degradation on others. ALPS (optimized on all 3 objectives) provides more balanced performance but offers only one single pruned model for all possible user preferences. MOSP enables users to select from various reasonable trade-offs, maintaining acceptable performance across all tasks while allowing preference-based customization.

Table 6: Comparison of MOSP and single-objective optimization using ALPS.

| Method | C4 | GSM8K | Code |
|---|---|---|---|
| ALPS (optimized on C4only) | **17.66** | 18.71 | 5.00 |
| ALPS (optimized on GSM8Konly) | 39.63 | **2.96** | 7.85 |
| ALPS (optimized on Codeonly) | 44.88 | 4.63 | **2.43** |
| ALPS (optimized on all 3 objectives) | 20.44 | 2.63 | 3.12 |
| MOSP ($\lambda = [1, 0, 0]$) | 18.80 | 3.37 | 3.20 |
| MOSP ($\lambda = [0, 1, 0]$) | 24.33 | 3.05 | 3.06 |
| MOSP ($\lambda = [0, 0, 1]$) | 25.82 | 3.43 | 2.53 |
| MOSP ($\lambda = [0.72, 0.14, 0.14]$) | 19.30 | 3.20 | 2.79 |

**Interpreting the multi-stage optimization.** For efficient Pareto front exploration, we design a multi-stage optimization which disentangles the shared and task-specific knowledge through the following interactions:

Stage 1 (Dual ADMM): Identifies a foundational "core support" representing weights broadly beneficial across all tasks. This captures common knowledge required for the multi-task problem and serves as a strong shared prior.

Stage 2 (Task-specific ADMM): Leverages the core support from Stage 1 to guide task-specific pruning. For each task, it performs a separate ADMM procedure where the core support acts as regularization, ensuring task-specific weights build upon the shared foundation while incorporating task specific adaptation.

This two-stage decomposition enables efficient Pareto front exploration, decoupling of shared knowledge preservation from task-specific optimization.

# G   Broader Impacts

This paper proposes a new method for providing personalized pruned models tailored to different users. By utilizing these pruned models, users can reduce computational costs, making the approach more environmentally friendly. However, as with other pruning methods, the models may produce more inaccurate outputs after pruning. Therefore, careful verification of the results is necessary during use.

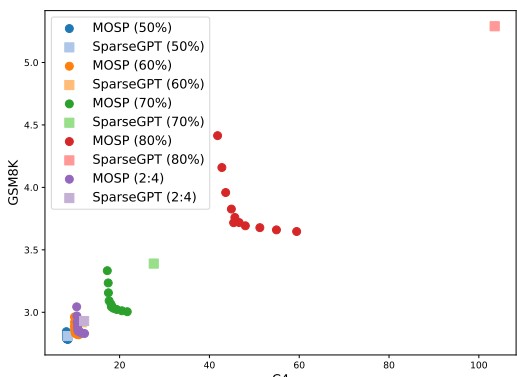

Figure 7: Test PPL on C4 and GSM8K for Llama-2-7B pruned to 50%, 60%, 70%, 80% unstructured sparsity and 2:4 semi-structured sparsity.

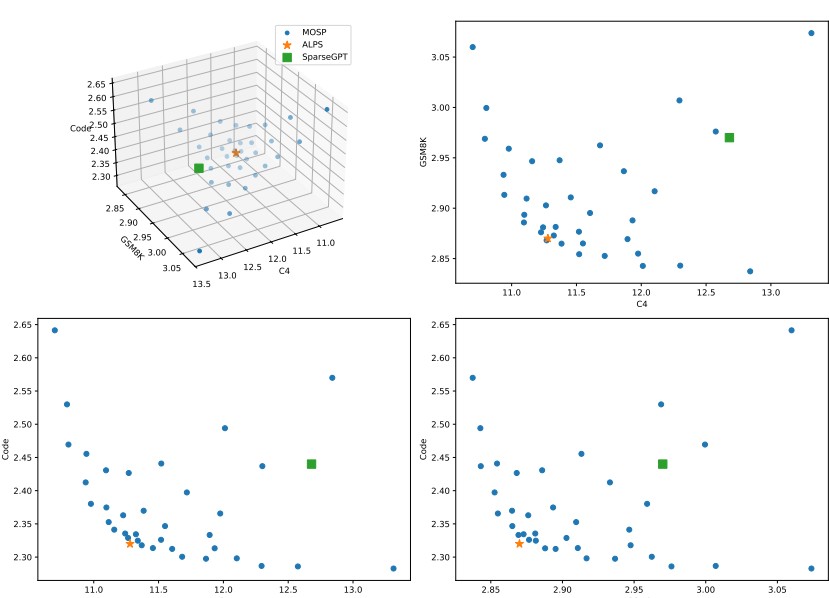

Figure 8: Test PPL on C4, GSM8K, and Code for Llama-2-7B pruned to 2:4 semi-structured sparsity.

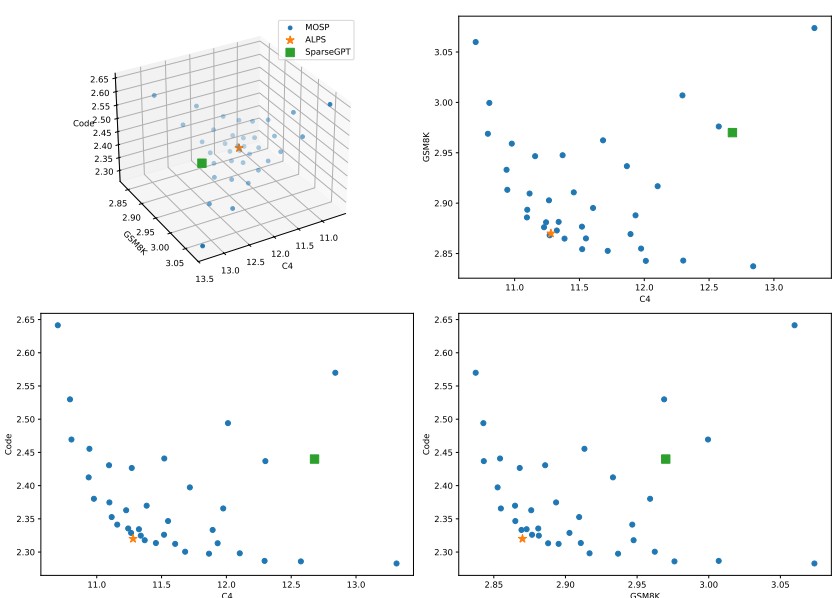

Figure 9: Test PPL on C4, GSM8K, and Code for Llama-2-7B pruned to 80% sparsity.

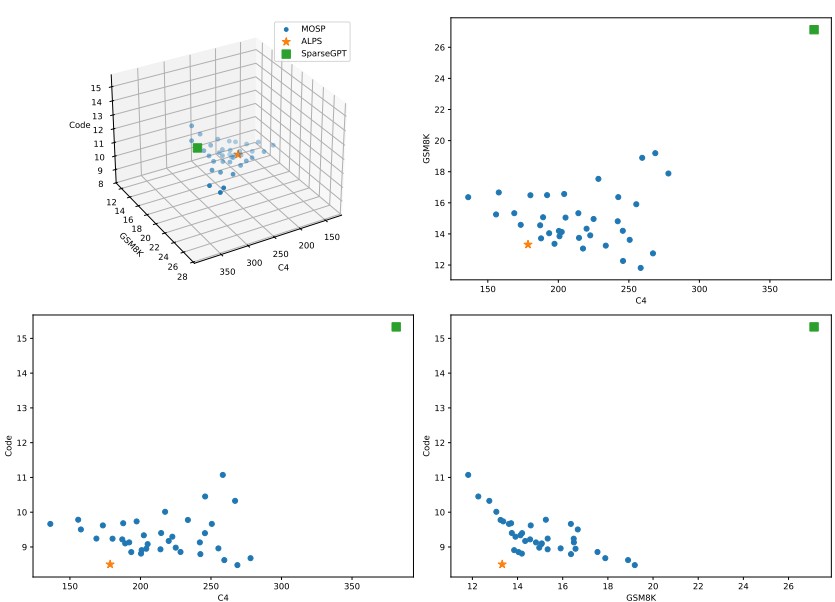

Figure 10: Test PPL on C4, GSM8K, and Code for OPT-1.3B pruned to 80% sparsity.

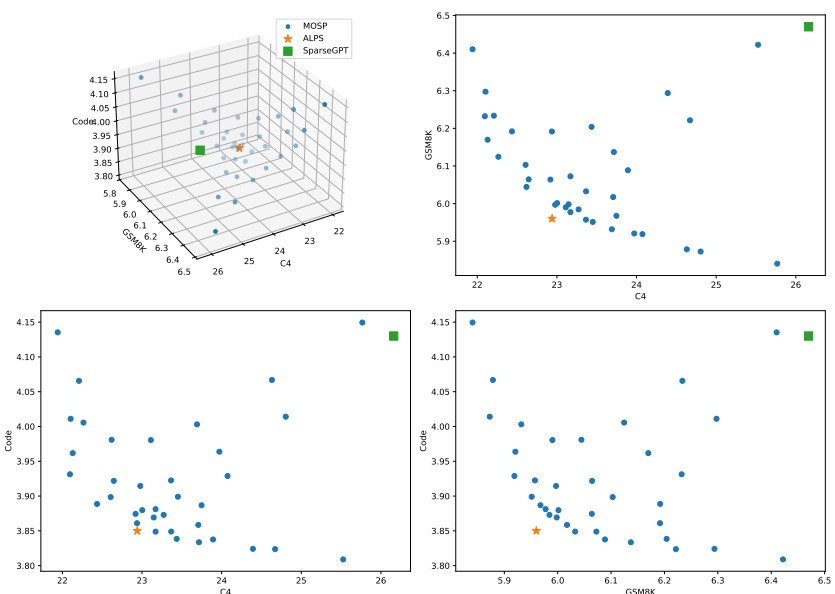

Figure 11: Test PPL on C4, GSM8K, and Code for OPT-2.7B pruned to 70% sparsity.

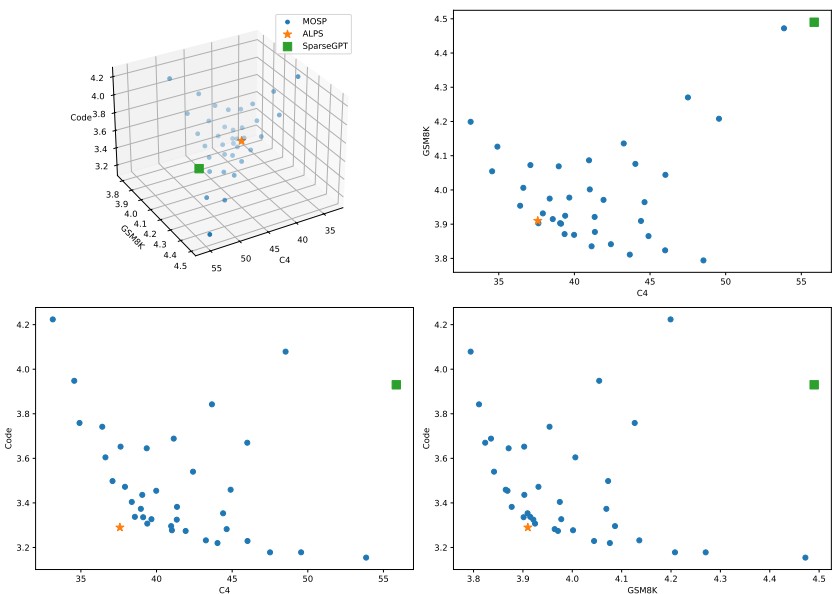

Figure 12: Test PPL on C4, GSM8K, and Code for LLaMMA-3-8B pruned to 70% sparsity.

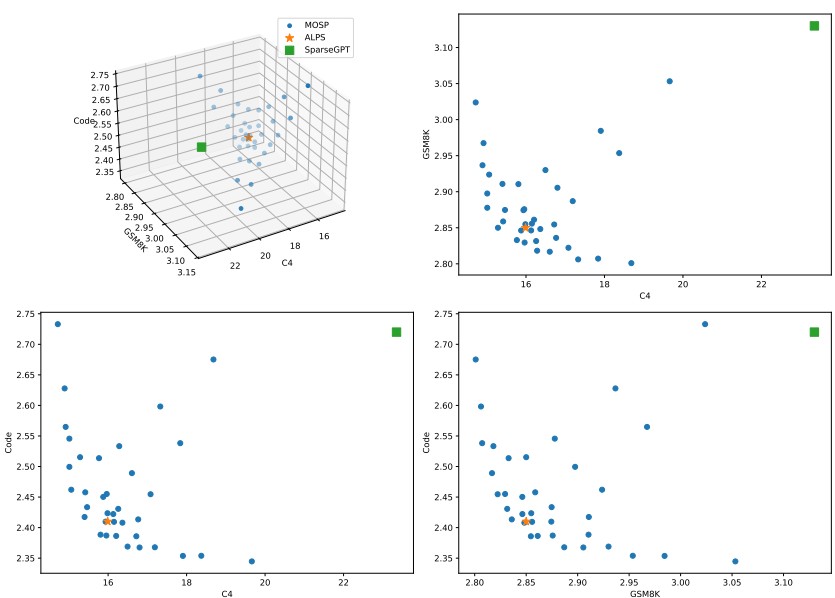

Figure 13: Test PPL on C4, GSM8K, and Code for LLaMMA-2-13B pruned to 70% sparsity.

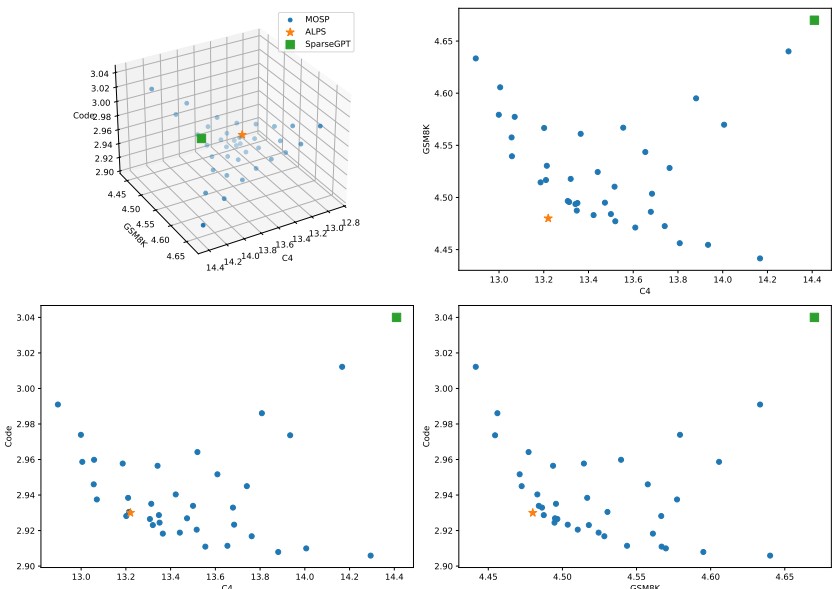

Figure 14: Test PPL on C4, GSM8K, and Code for OPT-30B pruned to 70% sparsity.

