# OpenReview forum: "Multi-Objective One-Shot Pruning for Large Language Models"
_NeurIPS.cc/2025/Conference — NeurIPS 2025 poster_

### Official Review · Reviewer_uTBb · 2025-06-17

**Clarity:** 4
**Significance:** 2
**Originality:** 4
**Rating:** 4
**Confidence:** 4

**Summary:**

LLMs are powerful but computationally expensive, and traditional one-shot pruning methods fail to account for their diverse applications. This paper proposes Multi-Objective One-Shot Pruning (MOSP), which frames pruning as a multi-objective optimization problem, producing a Pareto set of pruned models that balance different task performances.

**Questions:**

See Weaknesses

**Ethical Concerns:**

["NO or VERY MINOR ethics concerns only"]

**Final Justification:**

Experiments and responses are done well, which clarified their motivation of their work.
Overall, I think the paper is well written and I will keep my score advocating acceptance.

**Limitations:**

yes

**Quality:**

4

**Strengths And Weaknesses:**

# Strengths
- The manuscript is well written and easy to follow.
- Extensive experiments are conducted.
- Methods are well-defined and theoretically sound.

# Weaknesses
- My main concern is that the motivation of Multi-objective pruning is not fully justified. If "generality" is the key purpose, the necessity of the Pareto set of models is rather weak; on the other hand, if "specialization" is the goal, optimizing for a specific task will be better.
- That being said, in Figure 3, MOSP's overall performance gain over ALPS is marginal. Again, if we want a model that performs task A better, one may argue that it is better to optimize solely on A, rather than select a model from the Pareto set that performs task A a bit better than task B.
- Related to this point, it would be helpful to compare MOSP and other baselines against single-objective optimization results. This is not to say that MOSP should outperform the single-objective approaches; the single-objective results will reveal the upper bound for each task, which would be very informative for future research.
- Furthermore, this raises a fundamental question: to what extent should we consider two tasks as "distinct"? Are GSM8K and C4 truly heterogeneous tasks, and why should they form a Pareto frontier? -- Maybe, improving the Pareto frontier could have been a more valuable discussion.
- (minor) L154 : (3) --> (iii)

---

> ### Author Rebuttal · Authors · 2025-07-31
>
> Thank you very much for your positive feedback and insightful comments. We address each point below.
>
> ---
>
> **Q1:** My main concern is that the motivation of Multi-objective pruning is not fully justified. If "generality" is the key purpose, the necessity of the Pareto set of models is rather weak; on the other hand, if "specialization" is the goal, optimizing for a specific task will be better.
>
> **R1:**
> Thank you for your insightful comment. Our primary motivation is to serve the diverse range of user preferences that exist between the extremes of "pure generality" and various "pure specializations." For instance, different deployments of a model on edge devices may require different balances between capabilities: One user might prioritize language understanding (60\% importance) over coding (30\%) and math (10\%), while another may require strong mathematical reasoning (80\% importance).
>
> As the number of objectives increases, these preference combinations grow exponentially. MOSP efficiently addresses this by obtaining a set of trade-off models in a  **single pruning run**. This allows users to select the best-fit model for their specific needs post-training, **avoiding the significant computational cost of re-pruning for every new preference**. For instance, handling 100 distinct user preferences with traditional methods would take over 100 hours, whereas MOSP accomplishes this in just 1.36 hours. This approach also uniquely allows for dynamic preference adjustments. We will clarify this motivation more explicitly in the revised version.
>
> **Q2:** That being said, in Figure 3, MOSP's overall performance gain over ALPS is marginal. Again, if we want a model that performs task A better, one may argue that it is better to optimize solely on A, rather than select a model from the Pareto set that performs task A a bit better than task B.
>
> **R2:**
> Thanks for your comment. We believe the benefits of MOSP are significant. Recall that, as mentioned above, different users may have different preferences. Hence, as shown in Figure 3, when the user puts high priority on C4 (i.e., $\lambda = [1, 0]$), the test PPL of MOSP is 17.25 vs 18.17 of ALPS. This advantage increases with more objectives. For example, in the three-objective experiment
> (Table 3), when the user puts high priority on C4, the test PPL of MOSP is 18.80 vs 20.44 of ALPS. This advantage also increases with higher sparsity levels. For example, with 80\% sparsity, when the user puts high priority on C4, the test PPL of MOSP is 47.94 vs 53.88 of ALPS.
>
> Regarding optimize solely on A, we refer back to our reply **R1**: our method is designed for scenarios with diverse user preferences, many of which fall between rather than at the extremes. More specifically, as shown in our response **R3** below, optimizing solely for task A severely degrades performance on the other tasks, which is typically undesirable. Users generally prefer solutions that maintain reasonable performance across all tasks while emphasizing their priorities. We will clarify this in the revised version.
>
> **Q3:** Related to this point, it would be helpful to compare MOSP and other baselines against single-objective optimization results. This is not to say that MOSP should outperform the single-objective approaches; the single-objective results will reveal the upper bound for each task, which would be very informative for future research.
>
> **R3:** Thank you for this valuable suggestion. As suggested, we now add an experiment to compare MOSP against single-objective optimization using ALPS (i.e., running ALPS on each dataset separately). Note that this approach requires three separate pruning runs and cannot directly produce trade-off solutions by merging. The following table shows the test PPLs. For MOSP, we use four representative preferences.
>
> | Method                       | C4 | GSM8K | Code  |
> | ---------------------------- | :--------: | :----------: | :----------: |
> |ALPS (optimized on C4 only) |17.66 | 18.71| 5.00|
> |ALPS (optimized on GSM8K only) |39.63|2.96|7.85|
> | ALPS (optimized on Code  only) |44.88|4.63|2.43|
> | ALPS (optimized on all 3 objectives) |20.44|2.63|3.12 |
> | MOSP $\lambda = [1, 0, 0]$  | 18.80 | 3.37 | 3.20   |
> | MOSP $\lambda = [0, 1, 0]$  | 24.33 | 3.05 | 3.06|
> | MOSP $\lambda = [0, 0, 1]$ | 25.82 | 3.43 | 2.53  |
> |MOSP $\lambda = [0.72, 0.14, 0.14]$| 19.30 | 3.20 | 2.79|
>
> As can be seen, single-objective ALPS achieves optimal performance on its target task but shows significant degradation on others. ALPS (optimized on all 3 objectives) provides more balanced performance but offers only one single pruned model for all possible user preferences. MOSP enables users to select from various reasonable trade-offs, maintaining acceptable performance across all tasks while allowing preference-based customization.
>
> **Q4:** Furthermore, this raises a fundamental question: to what extent should we consider two tasks as "distinct"? Are GSM8K and C4 truly heterogeneous tasks, and why should they form a Pareto frontier? -- Maybe, improving the Pareto frontier could have been a more valuable discussion.
>
> **R4:** Thank you for this question. In multi-objective optimization (MOO), the various objectives (tasks in our context) do not need to be fully distinct. Indeed, in MOO, one is more concerned about how to balance the conflicts from the different objectives. If there is no conflict among the tasks, then all of them can be improved simultaneously. When there are conflicts (as is typically the case, even when the tasks are not completely distinct), improving one may lead to performance degradation of the other, and MOO addresses the problem of finding a good trade-off solution. Our experimental results also demonstrate this issue. As shown in the table above, the model jointly pruned using ALPS performs worse on each task compared to the models pruned individually for those tasks using ALPS. This performance degradation confirms that a trade-off exists between these objectives.  We will incorporate this clarification into the final version.
>
> **Q5:** (minor) L154 : (3) --> (iii)
>
> **R5:** Thank you for pointing this out. We will correct it and conduct a thorough proofread of the manuscript.
>
> Overall, we emphasize that one of MOSP's key contribution lies in obtaining a  set of
> trade-off solutions through a **single** efficient pruning run. These solutions effectively address the diverse range of user preferences between extremes, providing flexibility that traditional approaches cannot.
>
> ---
>
> We hope these responses have addressed your concerns. We are grateful for the valuable suggestions and welcome any further discussion.

---

> > ### Comment · Reviewer_uTBb · 2025-08-02
> >
> > Dear Authors,
> >
> > Thank you for the thoughtful responses to my questions and for the additional experiments.
> >
> > Also, thank you for clarifying the motivation - it makes more sense to me now.
> >
> > Overall, I think the paper is well written and I will keep my score advocating acceptance.
> >
> > Thank you.

---

### Official Review · Reviewer_UtLz · 2025-07-02

**Clarity:** 2
**Significance:** 3
**Originality:** 2
**Rating:** 4
**Confidence:** 4

**Summary:**

The authors present a novel formulation of the one-shot pruning problem for Large Language Models (LLMs) as a Multi-Objective Optimization (MOO) task, aiming to enhance generalization across multiple downstream tasks. The core methodology comprises a Dual ADMM framework for identifying shared core support and a Task-specific ADMM procedure for personalized pruning. The paper is further strengthened by rigorous theoretical convergence analysis and comprehensive empirical evaluations.

**Questions:**

See above

**Ethical Concerns:**

["NO or VERY MINOR ethics concerns only"]

**Final Justification:**

The rebuttal addressed my question effectively. I thus maintain my score of acceptance.

**Limitations:**

See above

**Paper Formatting Concerns:**

N

**Quality:**

3

**Strengths And Weaknesses:**

Strengths:

This method is more in line with the real user's differentiated preferences for multiple task requirements (e.g., language comprehension, mathematical reasoning, code generation).

Proposed MOSP (Multi-Objective One-Shot Pruning) framework, combining Dual ADMM and Task-specific ADMM, to achieve personalized models under user preferences while maintaining core weight sharing.

Theoretical proofs and experimental results are sufficient to further support the reliability of the method.

Weaknesses:

The introduction should give a slight description of the design of the dual ADMM, Task-spcific ADMM technique to make the main contributions more clear to the reader.

Is the fact that W^{prime} doesn't appear in the body of Eq. 4 caused by a typo?

It is difficult to distinguish between the MOSP method and other methods in Figure 4.

Methodology Phase 3: Refinement is not sufficiently described and it is suggested to expand on this in the appendix.

While the three-stage structure of MOSP is theoretically well-motivated, it lacks an interpretable explanation. The interaction between Dual ADMM and Task-specific ADMM is not clearly articulated, the structural relationship between core support and task-specific weights remains underexplored.

The experimental evaluation is limited to two- and three-objective scenarios, leaving the scalability of MOSP to higher-dimensional task settings (e.g., four or more objectives or finer-grained task divisions) unexplored. In addition, the approach lacks exploration of real-world human preferences, e.g., if the user requires that the model's language comprehension and code ability are equally important, should the hyperparameter be assigned 0.5 to both?

The paper does not provide unpruned baseline metrics, making it difficult to assess whether the results of the pruned model are directly usable.

The efficiency improvement of the model after pruning is suggested to provide some results for reference.

The ablation study lacks thoroughness in several aspects. (1) There is no comparison against the case where core support is entirely disabled (i.e., α = 0), making it difficult to assess its actual contribution. (2) The sensitivity of MOSP to different task combinations is not analyzed, which is important for understanding its generalization behavior. (3) The impact of varying the transformation parameter p in the λ-to-λ′ mapping is not systematically examined, leaving the effect of this design choice unclear.

---

> ### Author Rebuttal · Authors · 2025-07-31
>
> Thank you very much for your positive feedback and insightful comments. We address each point below.
>
> ---
>
> **Q1:** The introduction should give a slight description of the design of the dual ADMM, Task-spcific ADMM technique to make the main contributions more clear to the reader.
>
> **R1:**
> Thank you for this valuable suggestion. We will revise the introduction to provide clearer insight into our technical approach. Specifically, lines 35-37 will be revised as:
>
> "MOSP achieves this through a multi-stage process: First, we identify a common core support of weights crucial across all tasks using Dual ADMM optimization. This involves formulating the problem as bilevel optimization with ADMM applied to both inner and outer levels, with proven convergence guarantees. Second, using the identified supports, we perform a simplified ADMM for each task separately. This decoupling of shared knowledge preservation from task-specific optimization enables efficient on-the-fly generation of specialized sparse models based on user-defined preference vectors, allowing effective exploration of the Pareto front."
>
> **Q2:** Is the fact that $W^{\prime}$ doesn't appear in the body of Eq. 4 caused by a typo?
>
> **R2:** Yes, it is indeed a typo. In the inner optimization problem (4), all $\boldsymbol{W}$ on the right-hand side should be the optimization variable $\boldsymbol{W}'$. We will fix this in the final version. Thank you for catching this typo.
>
> **Q3:** It is difficult to distinguish between the MOSP method and other methods in Figure 4.
>
> **R3:** Figure 4 shows the Pareto fronts achieved by the proposed MOSP under different sparsity levels, and does not provide comparison with other baselines. We will add a new figure (similar to Figure 4) in the final version to compare MOSP with the baseline methods.
>
> **Q4:** Methodology Phase 3: Refinement is not sufficiently described and it is suggested to expand on this in the appendix.
>
> **R4:** Thank you for your suggestion. The refinement phase follows the standard refinement procedure in ALPS. As suggested, we will add a more complete description of it in the appendix in the final version.
>
> **Q5:** While the three-stage structure of MOSP is theoretically well-motivated, it lacks an interpretable explanation. The interaction between Dual ADMM and Task-specific ADMM is not clearly articulated, the structural relationship between core support and task-specific weights remains underexplored.
>
> **R5:** Thank you for this valuable suggestion. The multi-stage optimization disentangles the shared and task-specific knowledge through the following interactions:
>
> Stage 1 (Dual ADMM): Identifies a foundational "core support" ($W_c$) representing weights broadly beneficial across all tasks. This captures common knowledge required for the multi-task problem and serves as a strong shared prior.
>
> Stage 2 (Task-specific ADMM): Leverages the core support from Stage 1 to guide task-specific pruning. For each task, it performs a separate ADMM procedure where the core support acts as regularization, ensuring task-specific weights build upon the shared foundation while incorporating task specific adaptation.
>
> This two-stage decomposition enables efficient Pareto front exploration, decoupling of shared knowledge
> preservation from task-specific optimization. Due the rebuttal rules, we cannot include visualization results. We will include them to the final version, and will also incorporate this explanation into the methodology section.
>
> **Q6:** The experimental evaluation is limited to two- and three-objective scenarios, leaving the scalability of MOSP to higher-dimensional task settings (e.g., four or more objectives or finer-grained task divisions) unexplored. In addition, the approach lacks exploration of real-world human preferences, e.g., if the user requires that the model's language comprehension and code ability are equally important, should the hyperparameter be assigned 0.5 to both?
>
> **R6:**
> Thank you for this valuable suggestion. As suggested, we now add another objective, multilingual performance using ChineseWebText 2.0, to the three-objective experiment in Section 4.3 (with Llama-2-7B and 70\% sparsity). We use the following five preference vectors: $\lambda^{(1)} = [1, 0, 0, 0]$, $\lambda^{(2)} = [0, 1, 0, 0]$, $\lambda^{(3)} = [0, 0, 1, 0]$, $\lambda^{(4)} = [0, 0, 0, 1]$, $\lambda^{(5)} = [0.07, 0.07, 0.07, 0.80]$. The following table shows the testing PPL. As can be seen, MOSP can again identify the best trade-off solutions. Moreover, MOSP also achieves higher HV than ALPS (92.2 vs. 76.9).
>
>  | Method       | C4      |  GSM8K  | Code  | Chinese |
> | -------------| :-----: | :----: | :----: | :-----: |
> |Wanda|73.49 | 13.48| 11.38| 136.39|
> |SparseGPT | 30.52| 3.60 |  3.21| 16.57|
> |ALPS          |  20.57  |  3.17  |   2.67 | 9.53   |
> |MOSP($\lambda^{(1)} = [1, 0, 0, 0]$)|**19.17**| 3.42 |     3.37 | 14.31  |
> |MOSP($\lambda^{(2)} = [0, 1, 0, 0]$)|25.07    | **3.07**   | 3.25| 18.30  |
> |MOSP($\lambda^{(3)} = [0, 0, 1, 0]$)|26.69    | 3.50|   **2.57** | 17.31  |
> |MOSP($\lambda^{(4)} = [0, 0, 0, 1]$)|22.52    | 3.51   |  3.26  |**8.17**|
> |MOSP($\lambda^{(5)} = [0.07, 0.07, 0.07, 0.80]$)|20.19    |  3.34  |   2.97  | 8.52   |
>
> Due to time constraints in the rebuttal period, we present results for four objectives. In the final version, we will include more objectives (such as medical and legal domains) in the evaluation.
>
> Regarding human preferences, your interpretation is correct: the preference vector $\lambda$ directly reflects user-defined priorities. In other words, with only two objectives (language comprehension and coding) that are equally important, the user can set $\lambda = [0.5, 0.5]$. When language comprehension is more important, the user might use $\lambda = [0.8, 0.2]$. We will clarify this in the revised version.
>
> **Q7:** The paper does not provide unpruned baseline metrics, making it difficult to assess whether the results of the pruned model are directly usable.
>
> **R7:** Thank you for this suggestion. In response to your suggestion, we compare the proposed method with unpruned baseline on two dataset case (i.e., C4 and GSM8K). The unpruned Llama-2-7B baseline achieves test PPL of 6.97 on C4 and 2.73 on GSM8K, while our model pruned to 50\% sparsity achieves test PPL of 8.14 and 2.78, respectively.
> While some performance degradation is expected for pruned models (especially at high sparsity levels), our results show that MOSP provides superior Pareto-optimal solutions compared to existing methods, as shown in Section 4.2 and 4.3. We will inlcude the unpruned baseline results in the final version.
>
> **Q8:** The efficiency improvement of the model after pruning is suggested to provide some results for reference.
>
> **R8:** Thank you for this suggestion. For 2:4 sparsity, we observe ~1.5x speedup for linear layers on NVIDIA A6000. Due to the limited rebuttal period, we cannot provide timing results on other unstructured sparsity. As reference, SparseGPT [1] reports ~2.16x speedup at 60\% sparsity using the DeepSparse engine.  We will include more efficiency benchmarks in the final version.
>
> [1] Frantar & Alistarh, SparseGPT: Massive Language Models Can be Accurately Pruned in One-Shot, ICML 2023
>
> **Q9:** The ablation study lacks thoroughness in several aspects. (1) There is no comparison against the case where core support is entirely disabled (i.e., $\alpha$ = 0), making it difficult to assess its actual contribution. (2) The sensitivity of MOSP to different task combinations is not analyzed, which is important for understanding its generalization behavior. (3) The impact of varying the transformation parameter p in the $\lambda$-to-$\lambda'$ mapping is not systematically examined, leaving the effect of this design choice unclear.
>
> **R9:** Thank you for your comments. For the case where core support is entirely disabled (i.e., $\alpha$ = 0), indeed we have reported this experiment in Section 4.4 and Figure 6(a). As can be seen, setting $\alpha=0$ results in substantial performance degradation, confirming its critical contribution.
>
> For the sensitivity of MOSP to different task combinations, first, we have added an experiment with 4-objectives as mentioned in our reply **R6** above. Besides, we also experiment with the following 2-task combinations: (C4, Code) and (Code, Chinese). Empirically, MOSP successfully identifies the Pareto sets for these two combinations. However, due to rebuttal rules, we cannot upload the corresponding figures here, but they will be included in the final version.
>
> For the transformation parameter $p$, we have performed an ablation study in Section 4.4. As shown in Figure 6(b), models with extreme preferences (e.g., [1.0, 0]) perform similarly to those without transformation, while other models generally benefit from applying the transformation. During the rebuttal period, we have also examined two more $p$ values: $p=0.1$ and $p=1$. Due to rebuttal rules, we cannot upload the figure here, but it will be added to the final version.
>
> ---
>
> We hope these responses have addressed your concerns. We are grateful for the valuable suggestions and welcome any further discussion.

---

> > ### Comment · Reviewer_UtLz · 2025-08-05
> >
> > Your comment addressed my question effectively. I maintain my score of advocacy acceptance and look forward to seeing future versions of your paper.

---

### Official Review · Reviewer_yqiy · 2025-07-05

**Clarity:** 3
**Significance:** 3
**Originality:** 3
**Rating:** 4
**Confidence:** 3

**Summary:**

This paper introduces Multi-Objective One-Shot Pruning (MOSP), a method for efficiently reducing LLM size while considering multiple performance objectives simultaneously, such as general language understanding and math reasoning. The authors propose to generate a Pareto set of pruned LLMs with proof of convergence on the dual ADMM method. The authors validate the effectiveness of its method on various LLMs like LLama and OPT.

**Questions:**

none

**Ethical Concerns:**

["NO or VERY MINOR ethics concerns only"]

**Limitations:**

yes

**Quality:**

3

**Strengths And Weaknesses:**

### Strengthes:

1. The multi-objective formulation is novel. The authors propose to treat LLM pruning as multi-objective optimization, generating Pareto sets of models with different capability trade-offs rather than single solutions
2. The proposed method uses a shared core support as proven convergence, and enables efficient on-the-fly model generation with low computational overhead
3. Systematic evaluation on multiple model families and sparsity levels showing consistent improvements over state-of-the-art baselines



### Weaknesses:

1. Only three objectives are tested. The authors are expected to experiment on different objectives.
2. The paper only evaluates models with 7B+ parameters. Would this method work for smaller models, such as those with 3B parameters?

---

> ### Author Rebuttal · Authors · 2025-07-31
>
> Thank you very much for your positive feedback and insightful comments. We address each point below.
>
> ---
>
> **Q1:** Only three objectives are tested. The authors are expected to experiment on different objectives.
>
> **R1:**
> Thank you for this valuable suggestion. As suggested, we now add another objective, multilingual performance using ChineseWebText 2.0, to the three-objective experiment in Section 4.3 (with Llama-2-7B and 70\% sparsity). We use the following five preference vectors: $\lambda^{(1)} = [1, 0, 0, 0]$, $\lambda^{(2)} = [0, 1, 0, 0]$, $\lambda^{(3)} = [0, 0, 1, 0]$, $\lambda^{(4)} = [0, 0, 0, 1]$, $\lambda^{(5)} = [0.07, 0.07, 0.07, 0.8]$. The following table shows the testing PPL. As can be seen, MOSP can again identify the best trade-off solutions. Moreover, MOSP also achieves higher HV than ALPS (92.2 vs. 76.9).
>
>
>  | Method       | C4      |  GSM8K  | Code  | Chinese |
> | -------------| :-----: | :----: | :----: | :-----: |
> |Wanda|73.49 | 13.48| 11.38| 136.39|
> |SparseGPT | 30.52| 3.60 |  3.21| 16.57|
> |ALPS          |  20.57  |  3.17  |   2.67 | 9.53   |
> |MOSP($\lambda^{(1)} = [1, 0, 0, 0]$)|**19.17**| 3.42 |     3.37 | 14.31  |
> |MOSP($\lambda^{(2)} = [0, 1, 0, 0]$)|25.07    | **3.07**   | 3.25| 18.30  |
> |MOSP($\lambda^{(3)} = [0, 0, 1, 0]$)|26.69    | 3.50|   **2.57** | 17.31  |
> |MOSP($\lambda^{(4)} = [0, 0, 0, 1]$)|22.52    | 3.51   |  3.26  |**8.17**|
> |MOSP($\lambda^{(5)} = [0.07, 0.07, 0.07, 0.8]$)|20.19    |  3.34  |   2.97  | 8.52   |
>
> Due to time constraints in the rebuttal period, we present results for four objectives.
> In the final version, we will include more objectives (such as medical and legal domains) in the evaluation.
>
> **Q2:** The paper only evaluates models with 7B+ parameters. Would this method work for smaller models, such as those with 3B parameters?
>
> **R2:**
> Thank you for this question. Our method is indeed also effective for smaller models. As shown in Table 4 and Figure 9 (in the appendix), we have evaluated MOSP on OPT-2.7B, which demonstrates strong performance comparable to larger models. To further address your question, we now add an experiment on OPT-1.3B. With 70\% sparsity, MOSP achieves an HV value of 274.2, outperforming ALPS (211.8). Due to rebuttal rules, we cannot include figures here but will add these figures (similar to Figure 9) to the final version for a comprehensive evaluation across model sizes.
>
> ---
>
> We hope these responses have addressed your concerns. We are grateful for the valuable suggestions and welcome any further discussion.

---

> ### Comment · Reviewer_yqiy · 2025-08-09
>
> Thanks for the author's rebuttal, which addresses most of my concerns. After carefully reading the comment of other reviewers, I will keep my score.

---

### Official Review · Reviewer_PBt5 · 2025-07-10

**Clarity:** 3
**Significance:** 3
**Originality:** 3
**Rating:** 5
**Confidence:** 4

**Summary:**

The paper introduces Multi-Objective One-Shot Pruning (MOSP), a novel framework for pruning Large Language Models (LLMs) that optimizes for multiple objectives (e.g., general language understanding, mathematical reasoning, code generation) simultaneously. Unlike existing one-shot pruning methods (e.g., SparseGPT, ALPS), which optimize for a single objective, MOSP generates a Pareto set of pruned models representing different trade-offs.

**Questions:**

Na

**Ethical Concerns:**

["NO or VERY MINOR ethics concerns only"]

**Quality:**

3

**Strengths And Weaknesses:**

Strengths

The paper is well-written.

MOSP is the first to frame LLM pruning as a multi-objective optimization (MOO) problem, addressing the practical need for task-specific model customization.

Extensive experiments across 5 LLMs, 3 datasets (C4, GSM8K, Code), and diverse sparsity levels demonstrate MOSP’s superiority.



Weaknesses

The author should report the zero-shot or few-shot performance in the main text.

Are the pruning datasets the same as those of other methods? (Wanda, SparseGPT)?

---

> ### Author Rebuttal · Authors · 2025-07-31
>
> Thank you very much for your positive feedback and insightful comments. We address each point below.
>
> ---
>
> **Q1:** The author should report the zero-shot or few-shot performance in the main text.
>
>
> **R1:** Thank you for the suggestion. We now add a zero-shot experiment on the ARC-Challenge, ARC-Easy, and PIQA datasets as in ALPS. We use Llama-2-7B and 70\% sparsity. The experimental settings are the same as in Section 4.1 and those used for the experiment in Table 3. Since these are general language tasks with some reasoning components, we use the preference vector $\lambda = [0.72, 0.14, 0.14]$ (as in Table 3). The zero-shot accuracies are shown below.
>
> | Method | ARC-Challenge | ARC-Easy | PIQA |
> |----|------------|--------|----|
> | ALPS | 24.2 | 47.0 | 65.5 |
> | MOSP | **25.4** | **48.2** | **66.3** |
>
> As can be seen, MOSP shows better performance than ALPS. Due to time constraints during the rebuttal period, we only evaluate the above three datasets, more comprehensive zero-shot and few-shot evaluations will be performed in the final version of the paper.
>
> **Q2:** Are the pruning datasets the same as those of other methods? (Wanda, SparseGPT)?
>
> **R2:** Yes, we use exactly the same pruning datasets across all methods to ensure fair comparison.  We will further clarify it in the final version.
>
> ---
>
> We hope these responses have addressed your concerns. We are grateful for the valuable suggestions and welcome any further discussion.

---

### Note · Authors · 2025-08-12

We sincerely thank the reviewers and the Area Chair for their dedicated effort, positive feedback, and valuable suggestions.

We are encouraged that the reviewers recognized the importance of our work, i.e., the first to frame LLM one-shot pruning as a multi-objective optimization problem. As noted by the reviewers, our proposed MOSP method efficiently produces a Pareto set of pruned models in a single run, enabling users to select the optimal model for their specific needs without requiring re-pruning. We are also grateful for the acknowledgment of our extensive experiments and solid theoretical proofs.

In response to the reviewers' valuable suggestions, we have enhanced our paper by incorporating additional experiments (e.g., more objectives, smaller models, and zero-shot performance) and by providing more detailed motivation and technical explanations. We are pleased that these enhancements have successfully addressed the reviewers' concerns. All these improvements will be carefully incorporated into the final version and we will also add even more comprehensive experiments.

We sincerely hope that these efforts have further clarified the value of our work and would be grateful if the reviewers might kindly consider whether our paper merit stronger support in the final rating.

Thank you once again for your time, expertise, and constructive guidance throughout this review process.

---

### Decision · Program_Chairs · 2025-09-17

**Decision:**

Accept (poster)

**Comment:**

This paper introduces Multi-Objective One-Shot Pruning (MOSP), a framework for pruning LLMs by casting the problem as a multi-objective optimization task. Unlike existing one-shot pruning methods that focus on a single task, MOSP generates a Pareto set of models balancing multiple objectives (e.g., language understanding, reasoning, code), supported by a dual and task-specific ADMM framework with convergence guarantees. It provides a novel formulation and theoretical grounding of the proposed method, with systematic empirical evaluation. Reviewers also appreciated the clear motivation for handling diverse user preferences and the efficiency of generating multiple models in a single run.

However, some issues do exist: limited evaluation on small models and higher-dimensional objectives, and some clarity issues in the methodology description. The authors added more experimental results during the rebuttal, addressed most reviewer concerns. Therefore, despite some limitations, the balance of strengths and thorough rebuttal responses justify acceptance.